# Decoupling of interacting neuronal populations by time-shifted stimulation through spike-timing-dependent plasticity

**Mojtaba Madadi Asl**[1,2]*, **Alireza Valizadeh**[2,3]*, **Peter A. Tass**[4]

**1** School of Biological Sciences, Institute for Research in Fundamental Sciences (IPM), Tehran, Iran, **2** Pasargad Institute for Advanced Innovative Solutions (PIAIS), Tehran, Iran, **3** Department of Physics, Institute for Advanced Studies in Basic Sciences (IASBS), Zanjan, Iran, **4** Department of Neurosurgery, Stanford University School of Medicine, Stanford, CA, United States of America

* m.madadi@ipm.ir (MMA); valizade@iasbs.ac.ir (AV)

**Data Availability Statement:** All relevant data are within the paper and its Supporting information files. The simulation code is available at https://

## Abstract

The synaptic organization of the brain is constantly modified by activity-dependent synaptic plasticity. In several neurological disorders, abnormal neuronal activity and pathological synaptic connectivity may significantly impair normal brain function. Reorganization of neuronal circuits by therapeutic stimulation has the potential to restore normal brain dynamics. Increasing evidence suggests that the temporal stimulation pattern crucially determines the long-lasting therapeutic effects of stimulation. Here, we tested whether a specific pattern of brain stimulation can enable the suppression of pathologically strong inter-population synaptic connectivity through spike-timing-dependent plasticity (STDP). More specifically, we tested how introducing a time shift between stimuli delivered to two interacting populations of neurons can effectively decouple them. To that end, we first used a tractable model, i.e., two bidirectionally coupled leaky integrate-and-fire (LIF) neurons, to theoretically analyze the optimal range of stimulation frequency and time shift for decoupling. We then extended our results to two reciprocally connected neuronal populations (modules) where inter-population delayed connections were modified by STDP. As predicted by the theoretical results, appropriately time-shifted stimulation causes a decoupling of the two-module system through STDP, i.e., by unlearning pathologically strong synaptic interactions between the two populations. Based on the overall topology of the connections, the decoupling of the two modules, in turn, causes a desynchronization of the populations that outlasts the cessation of stimulation. Decoupling effects of the time-shifted stimulation can be realized by time-shifted burst stimulation as well as time-shifted continuous simulation. Our results provide insight into the further optimization of a variety of multichannel stimulation protocols aiming at a therapeutic reshaping of diseased brain networks.

## Author summary

To clinically advance different types of brain stimulation, e.g., deep brain stimulation or epicortical stimulation, in numerous clinical studies, typically only a few types of stimulus

github.com/MMadadiAsl/Stimulation-induced-decoupling.

**Funding:** PAT gratefully acknowledges funding support by the John A. Blume Foundation and the Foundation for OCD Research (New Venture Fund 011665-2020-08-01). The funders had no role in study design, data collection and analysis, decision to publish, or preparation of the manuscript.

**Competing interests:** I have read the journal's policy and the authors of this manuscript have the following competing interests: PAT works as consultant for Boston Scientific Neuromodulation and is inventor on a number of patents for invasive and non-invasive neuromodulation. The remaining authors declare that the research was conducted in the absence of any commercial or financial relationships that could be construed as a potential conflict of interest.

patterns have been delivered to different target areas in the midbrain or cortex. To further leverage the power of these clinical trials we here present a theoretical and numerical study demonstrating that the effects of brain stimulation may massively depend on variations of supposedly minor parameters. To this end, we introduce a time shift between stimulus trains delivered to two anatomically separate neuronal populations interacting through plastic synapses. Depending on the specific time shift, stimulation may be ineffective or induce pronounced changes of the connections between and, in turn, within the neuronal populations, ultimately causing a long-lasting unlearning of abnormal neuronal synchrony. To thoroughly understand the time shift-induced decoupling mechanism, we first consider a simple two-neuron motif of two leaky integrate-and-fire neurons. Intriguingly, our results obtained in the two-neuron motif are in excellent agreement with the two-population scenario, illustrating the predictive power of comparably simple models. Our results are important in the context of the control of plastic neuronal networks and provide testable hypotheses for the improvement of clinically used stimulation techniques.

## Introduction

Synaptic connections in cortical networks are highly adaptive due to activity-dependent synaptic plasticity [1]. Reshaping of connectivity patterns by plasticity mechanisms based on external stimuli is necessary for normal brain function such as appropriate motor actions [2, 3]. Spike-timing-dependent plasticity (STDP) provides a mechanistic model for the modification of the strength of synaptic connections according to the temporal coincidence of pre- and postsynaptic activity [4–7]: Synapses are strengthened when the presynaptic spike precedes the postsynaptic spike, whereas they are weakened in the reverse scenario [5]. In this way, neuronal activity shapes the synaptic organization of brain networks [1, 8–11] which, in turn, adjusts the activity of neurons in a feedback loop [12–15]. Normal brain function is shaped by structurally and functionally interconnected brain areas mediated by plastic, network-based structure-function relationships [16].

Several brain disorders such as Parkinson's disease (PD) [17–21], essential tremor [22–24], Alzheimer's disease (AD) [25, 26], epilepsy [27–30], schizophrenia [31] and autism spectrum disorder (ASD) [32] are linked with abnormal brain activity and connectivity. For instance, network interactions are severely affected in PD patients due to increased connectivity in cortical/subcortical regions [19, 20], that are associated with abnormal neuronal activity [33–35]. Not all neurons and synapses are involved in the pathology to the same extent. For instance, in PD, pallido-subthalamic gamma-aminobutyric acid (GABA)ergic synapses are up-regulated [19], whereas glutamatergic cortico-subthalamic are down-regulated [36]. Abnormal synaptic connectivity may induce abnormally increased neuronal synchrony, as in PD [19, 37], or abnormally reduced neuronal synchrony, e.g., in AD [25, 26, 38], likely due to aberrant synaptic plasticity [37, 39–41]. Furthermore, increased long-range brain connectivity may underlie seizure facilitation in patients with epilepsy [30] and may predict symptom severity in ASD [32].

Therefore, counteracting abnormal changes in brain activity and connectivity by therapeutic interventions may provide effective treatment approaches for brain disorders. Neural circuits in the brain can be modulated by a variety of invasive and noninvasive electrical stimulation strategies aiming at the recovery of normal circuit functions [42–46]. For instance, noninvasive stimulation of cortical regions can be realized by transcranial direct current

stimulation (tDCS) or transcranial alternating current stimulation (tACS) which have shown promising effects on motor and cognitive functions in neuropsychiatric disorders [43, 45, 47]. On the other hand, invasive high-frequency ($> 100$ Hz) deep brain stimulation (HF-DBS) is an effective clinical therapy for pathological conditions such as medically refractory PD and epilepsy [42, 48, 49]. However, reappearance of symptoms soon after the discontinuation of stimulation [50] entails chronic stimulation which may further side effects [51, 52]. Also, DBS delivered to the subthalamic nucleus (STN) and globus pallidus internus (GPi) is considered as ineffective for treating impairment of gait and balance and is little beneficial for or even worsens speech impairment [53]. Based on computational studies, it was suggested to counter-act abnormal neuronal synchrony by stimulation techniques designed to specifically cause desynchronization [54, 55].

In computational studies in oscillator and neuronal networks, different scenarios have been proposed for the desynchronization of neural populations by single-site stimulation (i.e., targeting one population) or multi-site stimulation (i.e., targeting two or more populations). For instance, desynchronization may be realized by demand-controlled delayed feedback stimulation [56–59] where the whole network is stimulated and registered at the same time which can be challenging in a clinical situation. This can be resolved by spatially splitting the whole population into two separate subpopulations, one being stimulated and the other being measured [60]. A more efficient approach has been suggested to separate the stimulation and registration processes in time rather than in space [61], i.e., by time-delayed feedback control of the pathological activity. However, these methods require a real-time measurement of the network activity. More importantly, smooth, non-pulsatile feedback stimulation techniques typically violate safety requirements, in particular, charge density limits [62–64]. Hence, based on computational studies, it was suggested to use linear and nonlinear delayed feedback signals to modulate amplitude and sign of continuous pulse train stimulation [65–67]. Alternatively, using multiple sites and tuning the temporal pattern of stimulation may have huge consequences on the stimulation outcome [68–70]. For instance, coordinated reset (CR) [71] stimulation is a theory-based multichannel patterned stimulation that targets subpopulation of neurons at different sites sequentially, i.e., in a timely coordinated manner [71, 72]. Computationally, CR stimulation can shift network dynamics from pathologically synchronized states to more physiologically favored states with desynchronized activity and, hence, induce long-lasting desynchronizing effects that outlast stimulation offset [72]. The desynchronizing effects [71], cumulative effects [73] and long-lasting effects [72] of the CR stimulation were validated in pre-clinical as well as clinical proof-of-concept studies [74–78].

Prolonged stimulation effects are desirable since they can induce sustained therapeutic outcome that outlast the cessation of stimulation. However, desynchronizing stimulation may not necessarily weaken pathologically strong synapses between the neurons during stimulation [79, 80]. Rather, long-lasting desynchronizing effects can be realized by decoupling neurons [80], i.e., by desynchronizing the overly synchronized activity of neurons and, furthermore, reduction of the pathologically strong synaptic connections between neurons. Overly synchronized neuronal activity can arise due to either strong local connections within the brain regions or due to the excessively potentiated long-range connections between the different brain regions [81–84]. Therefore, therapeutic stimulation techniques aiming at the reduction of abnormally synchronized activity could target both local connections and long-range projections.

In the normal brain, information exchange between functionally specialized regions is crucial for higher order brain functions [82, 85]. Such an inter-areal communication takes place through long-range synaptic connections across the brain [86–88] and phase synchronization may allow for controlling the inter-areal information exchange by dividing the time into the

windows of high and low excitability [85]. Coherence between the activities of brain areas might arise from generic mechanisms across cortex determined by long-range excitatory projections [88]. Abnormal long-range connectivity between brain areas was proposed as a potential pathophysiological mechanism [87, 89]. Pathological changes in inter-areal connectivity can cause a functional reorganization of the brain characterized by altered activity and functional connectivity patterns and, thereby impairs the normal brain function [87, 89, 90].

Cortical stimulation is one of the therapeutic strategies that typically requires a less invasive surgical procedure in comparison to DBS surgery. Experimentally, it was shown that cortical stimulation can induce changes in synaptic connectivity between two interacting networks [91–95], in this way restoring relevant features of physiological connectivity. As shown both computationally and experimentally [95, 96], rewiring of synaptic connectivity between two interacting populations can be realized, e.g., by applying a specific set of stimuli to two populations to induce a time shift (delay) between their activity so that inter-population synapses could be regulated through STDP. This motivated us to study a stimulation paradigm applied to a cortical network model with more realistic assumptions where we explicitly considered transmission delays along with STDP. We hypothesized that appropriate temporal detuning of two stimulus trains delivered to two different populations, e.g., a time shift between stimuli, could effectively decouple the two populations through STDP and induced pronounced effects outlasting the cessation of stimulation. In fact, one of the goals of this computational study is to demonstrate how minor parameter changes in comparably simple stimulation protocols may massively change stimulation outcome.

To test our hypothesis, we used a generic cortical model to study theoretical conditions for decoupling two initially strongly coupled neuronal populations (modules) by repetitive stimulation of both populations with a time shift. To provide a theoretical basis for the simulation results, we first considered a reciprocally coupled two-neuron motif with plastic synapses and theoretically analyzed the optimal range of stimulation frequency and time shift to decouple neurons with a given set of STDP parameters and transmission delays. We then simulated a two-module model of cortical networks composed of weakly coupled excitatory and inhibitory neurons within each population. The two populations interacted via inter-population excitatory-to-excitatory delayed connections modified by STDP. In this study we focused on the modification of inter-population synapses between the two modules to study the effect of the time-shifted two-site stimulation on the evolution of the plastic synapses between the two populations and the resultant effect on the dynamics of the network. To this end we assumed that the local connections within each population are weak and static. These assumptions meant that the synchronized activity of the network was due to the pathologically strong long-range connections and, furthermore, helped us to capture the pure effect of the modification of the long-range connections on the reduction of synchronized oscillatory activity of the network. Due to the initially strong inter-population synapses, the initial activity of the modules mimicked a pathological condition characterized by high firing rates of the neurons and large-amplitude collective oscillations. Stimuli were separately delivered to the two modules, each of them affecting an entire module, respectively. A dedicated time shift between the stimulus trains for the two modules modified inter-population synapses through STDP. We showed that the time-shifted stimulation enables the network to unlearn pathologically strong synaptic interactions between the two modules. Effective decoupling of the two modules ultimately caused a desynchronization of the network that persisted after stimulation cessation.

We showed that the time shift scheme may work in a rather generic manner. It can be realized by time-shifted trains of single stimulus pulses as well as patterned delivery of time-shifted bursts. The STDP potentiation and depression rates and time constants, and the delay in the transmission of signals along the inter-population connections determine the ultimate

stimulation-induced distribution of the synaptic strengths after stimulation offset. Our generic results may contribute to the further development of temporally patterned stimulation in a variety of multichannel stimulation protocols optimized for unlearning pathological connectivity between neurons which can be adapted for cortical stimulation to induce long-lasting therapeutic effects by shifting the dynamics of the diseased brain towards healthy attractor states.

## Methods

### Neuron and network model

The pairwise analysis was performed on a two-neuron motif comprising two excitatory leaky integrate-and-fire (LIF) neurons [97] connected by reciprocal plastic synapses (Fig 1A). As a

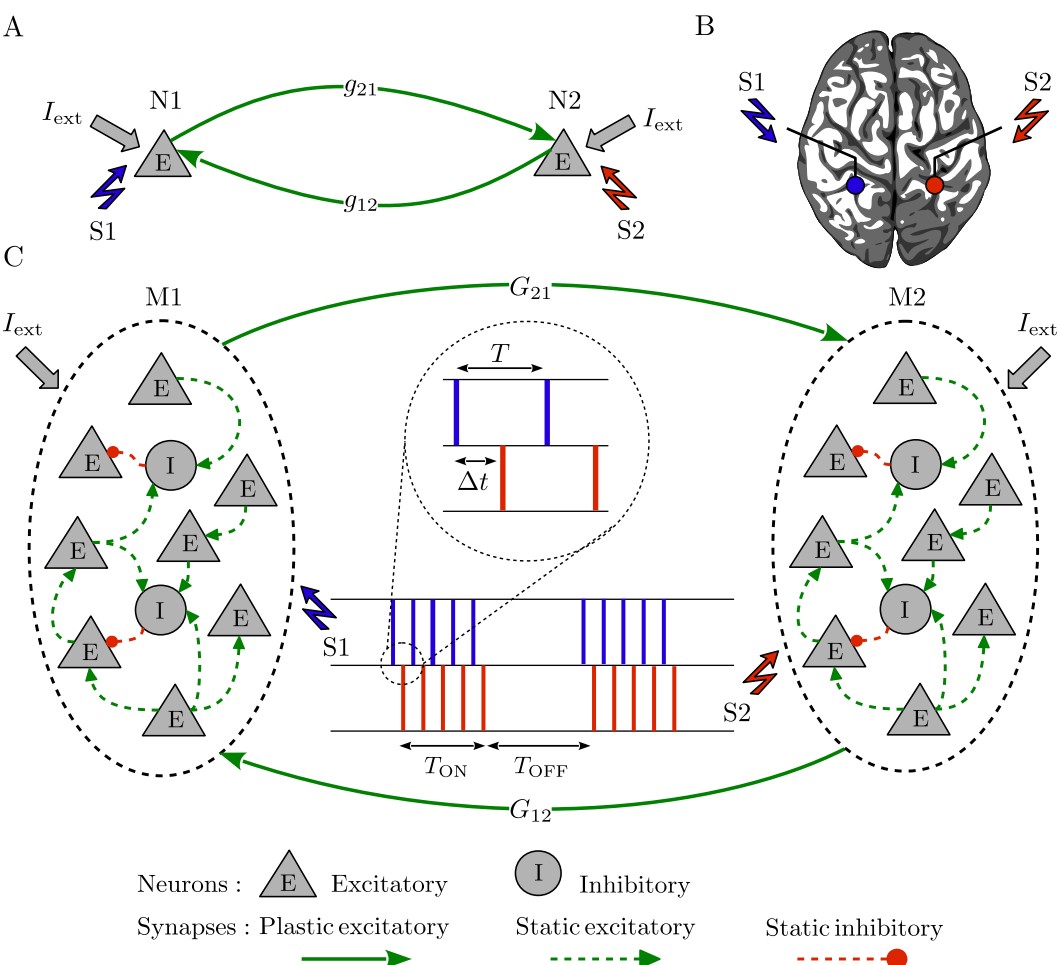

**Fig 1. Representation of the two-neuron motif and neuronal network.** (**A**) Two excitatory neurons coupled by reciprocal plastic synapses with strength $g_{21}/g_{12}$. (**B**) Schematic of the brain and the stimulation electrodes (circles) where the two stimulation signals (S1/S2) were separately delivered to two representative populations in panel C, e.g., in two hemispheres. (**C**) Two reciprocally connected neuronal populations (modules) characterized by inter-population excitatory-to-excitatory plastic synapses with mean coupling $G_{21}/G_{12}$. Each stimulation signal consists of intermittent bursts where $T$ represents inter-pulse interval within a burst and $\Delta t$ is the time shift between the two stimulation signals delivered to each module. $T_{ON}$ is the stimulation ON-epoch for each stimulation burst and $T_{OFF}$ is the stimulation OFF-epoch between two successive bursts within each stimulation signal.

model of cortical neuronal networks [81, 84, 98], two populations (i.e., modules), e.g., in two hemispheres (see Fig 1B), were connected via inter-population plastic excitatory-to-excitatory synapses (Fig 1C) [86, 99, 100]. Each module is a local balanced network (see S1(A) Fig) that consists of $N = 200$ LIF neurons randomly connected in a sparse manner [101], of which $N_{ex} = 0.8N$ were excitatory and $N_{in} = 0.2N$ were inhibitory in 4:1 proportion. The strength of inhibitory synapses within each module was on average 4-fold the strength of excitatory synapses (see S2A and S2B Fig), ensuring local excitation-inhibition balance [102, 103].

In the normalized units, subthreshold dynamics of the dimensionless membrane potential, i.e., $v_i(t) = V_i(t)/V_{th}$, of neuron $i$ is described by the following differential equation:

$$\frac{dv_i(t)}{d\lambda} = -v_i(t) + I_{syn}(t) + I_{ext}(t) + I_{stim}(t), \tag{1}$$

where $\lambda = t/\tau_m$ is the dimensionless time in the units of the membrane time constant $\tau_m = 10$ ms. When $v_i(t)$ reaches the firing threshold $v_{th} = 1$, the neuron fires and the membrane potential resets to the resting value $v_r = 0$. $I_{syn}(t) = \sum_j I_{ij}^{nn}(t) + \sum_j I_{ij}^{nm}(t)$ is the synaptic current, where $I_{ij}^{nm}(t) = c_{ij}^{nm} g_{ij}^{nm} s_j^{nm}(t - \tau)$ is the intra- ($n = m$) or inter-population ($n \neq m$) synaptic current from the presynaptic neuron $j$ in module $m$ to the postsynaptic neuron $i$ in module $n$. $c_{ij}^{nm}$ and $g_{ij}^{nm}$ are the corresponding elements of the adjacency (**C**) and synaptic strength (**G**) matrices, respectively. $\tau = \tau_d + \tau_a$ is the total inter-module transmission delay in forward or backward direction, i.e., the sum of dendritic($\tau_d$) and axonal ($\tau_a$) transmission delays in the synapse connecting the pre- and postsynaptic neurons. $s_j^{nm}(t) = \sum_f \exp((t - t_j^{(f)})/\tau_s)\,\Theta(t - t_j^{(f)})$ denotes the spiking activity of neuron $j$ with a time constant $\tau_s = 5$ ms [104], where $t^{(f)}$ is the firing time of neurons and $\Theta(t)$ is the Heaviside step function. Neuronal, synaptic and network model parameters are given in Table 1.

**Table 1. Neuronal, synaptic and network model parameters [84].**

| Parameter | Symbol | Value |
|---|---|---|
| Membrane time constant | $\tau_m$ | 10 ms |
| Spiking threshold | $V_{th}$ | −40 mV |
| Resting membrane potential | $V_r$ | −60 mV |
| Synaptic time constant | $\tau_s$ | 5 ms |
| Intra-module transmission delay | $\tau_{intra}$ | 0.0 ms |
| Total inter-module transmission delay | $\tau_{inter}$ * | 11 ms |
| Upper bound of the synaptic strengths | $g_{max}$ | 1.00 |
| Lower bound of the synaptic strengths | $g_{min}$ | 0.05 |
| Intra-module inhibitory mean synaptic strength | $g_{intra}^{in}$ | 0.8 |
| Intra-module excitatory mean synaptic strength | $g_{intra}^{ex}$ | 0.2 |
| Inter-module excitatory mean synaptic strength | $g_{inter}^{ex}$ | 0.8 |
| Total number of neurons per module | $N$ | 200 |
| Number of excitatory neurons per module | $N_{ex}$ | 160 |
| Number of inhibitory neurons per module | $N_{in}$ | 40 |
| Intra-module connection probability | $p_{intra}$ | 0.15 |
| Inter-module connection probability | $p_{inter}$ | 0.15 |

*Total inter-module delay is given by $\tau_{inter} = \tau_d + \tau_a$, where the dendritic delay was fixed at $\tau_d = 0.5$ ms and the remaining delays were assigned to the axonal delay $\tau_a$.

## External background input

The noisy input, $I_{\text{ext}}(t)$, represents the external input from other parts of the brain that are not considered in the model, considering the high spiking variability observed in cortex [105], modeled as a homogeneous Poisson process which best represents single neuron dynamics [81, 84]. In the model, each excitatory neuron within each module was driven by 8000 independent Poisson excitatory spike trains, each with a mean rate of 1 spike/s. Each inhibitory neuron within each module was driven by 6500 independent Poisson excitatory spike trains, at the same mean rate. The synaptic strengths, transmission delays and external input were tuned such that each population (when isolated) operated in an inhibition-stabilized regime, characterized by irregular individual firing of neurons and desynchronized network activity (see S1 (B) Fig) [84, 102, 104]. However, the network operating point was close enough to the oscillatory regime such that strong long-range excitation could push the global activity towards a synchronized regime [84, 86, 99]. The intra-population inhibitory synaptic strengths were picked from a Gaussian distribution with mean 0.8 and standard deviation 0.05, whereas the intra-population excitatory synaptic strengths were picked from a Gaussian distribution with mean 0.2 and the same standard deviation (see S2A and S2B Fig). The inter-population excitatory synaptic strengths were chosen from a Gaussian distribution with mean 0.2 and standard deviation 0.05 (see S1(C) Fig).

## Stimulation protocol

$I_{\text{stim}}(t)$ in Eq (1) represents the stimulation current (S1/S2 in Fig 1) composed of intermittent bursts that are simultaneously delivered to all (excitatory and inhibitory) neurons embedded within a population:

$$I_{\text{stim}}(t) = K \sum_i \delta(t - t_{\text{stim}}^{(i)}), \qquad (2)$$

where $K$ is a dimensionless parameter representing the stimulation intensity, $t_{\text{stim}}^{(i)}$ denotes the onset time of the individual stimulation pulses and $\delta(t)$ is the Dirac delta function. Given the initial point of the stimulation ($t_{\text{stim}}^{(1)}$), the time of the next pulse onset is determined by the following stimulation protocol:

$$t_{\text{stim}}^{(i+1)} = t_{\text{stim}}^{(i)} + T + \delta(k - i)(T_{\text{OFF}} - T), \qquad (3)$$

where $T$ is the inter-pulse interval which alternatively represents the intra-burst frequency, i.e., $\nu = 1/T$. The total duration of stimulation, i.e., *stimulation epoch*, was $T_{\text{stim}} = 5$ s. $k = 5$ is the number of pulses within a burst delivered for the duration of ON-epoch ($T_{\text{ON}} = 120$ ms). $T_{\text{OFF}} = 360$ ms represents the stimulation OFF-epoch between two successive stimulation bursts, i.e., the inverse of burst delivery rate, within each stimulation signal (see Fig 1C).

S1/S2 stimulation signals could separately target two neurons (N1/N2 in Fig 1A) or two representative populations (M1/M2 in Fig 1C) in two hemispheres as shown in Fig 1B. Assuming that the S1 stimulation signal is delivered to N1 (M1) at $t_{\text{stim}}^{(1)}(\text{S1}) = 10$ s, the S2 stimulation signal is delivered to N2 (M2) with a time shift $\Delta t = t_{\text{stim}}^{(1)}(\text{S2}) - t_{\text{stim}}^{(1)}(\text{S1})$ that represents the time shift between the onset times of stimulations delivered to the two neurons (populations), as measured between the first pulse in each train.

## Spike-timing-dependent plasticity (STDP)

Pre- ($j$) and postsynaptic ($i$) neurons within each module were connected to each other via instantaneous static synapses with strength $g_{ij}^{nn}$, whereas the two modules were connected by

**Table 2. Plasticity model parameters [7, 107, 108].**

| Parameter | Symbol | Value |
|---|---|---|
| Maximum potentiation amplitude | $A_+$ | 0.008 |
| Maximum depression amplitude | $A_-$ | 0.005 |
| Potentiation time constant | $\tau_+$ | 10 ms |
| Depression time constant | $\tau_-$ | 20 ms |

inter-module excitatory-to-excitatory plastic synapses with transmission delays from the pre-synaptic neuron $j$ in module $m$ to the postsynaptic neuron $i$ in module $n$ with strength $g_{ij}^{nm}$ modified according to the following pair-based STDP rule [6]:

$$\Delta g_{ij}^{nm} = A_\pm \, \text{sgn}(\Delta t + \xi) \exp(-|\Delta t + \xi|/\tau_\pm), \tag{4}$$

where $A_\pm$ and $\tau_\pm$ are the learning rate and the effective time constant of synaptic potentiation (upper) and depression (lower sign), respectively, and $\text{sgn}(\Delta t)$ is the sign function. $\Delta t = t_{\text{post}} - t_{\text{pre}}$ is the instantaneous time lag between pre- and postsynaptic spike pairs and $\xi = \tau_d - \tau_a$ indicates the effective delay perceived at the synapse, i.e., the difference between dendritic and axonal transmission delays [106].

Evaluated over the time interval between two successive spikes ($T$), the potentiation and depression terms in Eq (4) compete to determine the net synaptic change in a synapse as follows [106]:

$$\Delta g_{ij}^{(T)} = A_+ \exp(-|\Delta t^+|/\tau_+) - A_- \exp(-|\Delta t^-|/\tau_-), \tag{5}$$

assuming that $|\Delta t^+| = \Delta t + \xi$ is the time lag used by the STDP rule for potentiation of the synapse and $|\Delta t^-| = T - |\Delta t + \xi|$ is the depression time lag.

We used a generic, dimensionless model within which the synaptic weights are scaled so that they reproduce the dynamics that mimic those observed in normal and diseased brains in a realistic situation. The synaptic strengths were updated by an additive rule at each step of the simulation, $g \rightarrow g + \Delta g$. The value of the synaptic strengths was restricted in the range $[g_{\text{min}}, g_{\text{max}}] \in [0.05, 1.00]$. The synaptic strengths were set to $g_{\text{min}}$ ($g_{\text{max}}$) via hard bound saturation constraint once they crossed the lower (upper) bound of their allowed range. Plasticity model parameters are given in Table 2.

## Data analysis

**Inter-population mean coupling.** The mean coupling strength between the modules from the presynaptic module $m$ to the postsynaptic module $n$ was measured by calculating the average of the plastic inter-population synaptic strengths at a given time:

$$G_{nm}(t) = \frac{1}{N} \sum_{i=1}^{N} \sum_{j=1}^{N} c_{ij}^{nm} g_{ij}^{nm}, \tag{6}$$

where $N$ is the total number of neurons in each module. Furthermore, the time averaged inter-population mean coupling between the two modules in the network ($G_{\text{ave}}$) was evaluated over 10 s of network activity after stimulation offset.

**Population activity.** The population activity of module $i$ in the network was calculated by counting the number of spikes in a time interval which gives the number of active neurons at

that interval:

$$A_i(t) = \frac{1}{N} \sum_{j=1}^{N} \sum_{f} \delta(t - t_j^{(f)}),$$ (7)

where $N$ is the total number of neurons in module $i$ and $t^{(f)}$ is the firing time of individual neurons.

**Pairwise correlations.** The Pearson correlation coefficient was used to calculate the spike count correlation between pairs of neurons $(i, j)$ in each module that is given by:

$$r_{ij} = \frac{c_{ij}}{\sigma_i \sigma_j},$$ (8)

where $c_{ij}$ is the covariance between spike counts of the two neurons calculated from their spike trains, and $\sigma_i$ is the standard deviation of spike time distribution given by the corresponding spike train. Correlations were calculated based on the spike times resulted from 10 s of network activity before/after stimulation on/offset.

**Spike count irregularity.** The coefficient of variation of the inter-spike intervals (ISIs) was calculated as a measure of the irregularity of spiking activity of neuron $i$:

$$CV_i = \frac{\sigma_i}{\mu_i},$$ (9)

where $\sigma_i$ is the standard deviation and $\mu_i$ is the mean of the ISIs calculated from the spike time distribution of neuron $i$ given by its spike train evaluated over 10 s of network activity before/after stimulation on/offset.

**Population Fano factor.** The population Fano factor (pFF) was used to measure the synchrony of population activity in module $i$, defined as [109]:

$$pFF_i = \frac{\sigma^2[A_i(t)]}{\mu[A_i(t)]},$$ (10)

where $A_i(t)$ represents the population activity defined in Eq (7), and $\sigma^2$ and $\mu$ are the variance and mean of the population activity, respectively. The pFF evaluates the normalized amplitude of the variation of the population activity which increases when the neurons fire in synchrony [84, 110]. Smaller values of the pFF correspond to desynchronized states, whereas greater values of the pFF imply synchrony in the network. The pFF was evaluated over 10 s of network activity after stimulation offset.

## Results

### Two-neuron motif

We first studied if the time-shifted stimulation can decouple a two-neuron motif, i.e., reduce the initially strong synaptic connections and, furthermore, suppress the firing activity of neurons. For this, we explored the evolution of the synaptic strengths in a motif consisted of two stochastically firing excitatory neurons coupled by reciprocal plastic synapses (see Fig 1A). This motif is representative of two neurons in two different populations connected by long-range connections and as will be shown below by demonstrating that results obtained in the motif can adequately predict those for two connected populations. For simplicity and for analytical tractability, we considered the classical STDP profile characterized by asymmetric modification of the synapses based on Eq (4) [6]. We set the parameters of STDP according to the biologically realistic, generic forms of cortical STDP profiles [7] characterized by larger

potentiation rate (i.e., $A_+ > A_-$) and relatively longer time window for synaptic depression (i.e., $\tau_+ < \tau_-$). The temporal STDP parameters are consistent with experimental observations, e.g., in rat cortical slices [7]: $\tau_+ = 13$ ms and $\tau_- = 34$ ms, or those originally reported in rat hippocampal cultures [6]: $\tau_+ = 19$ ms and $\tau_- = 34$ ms. Based on these observations, the direction of the synaptic change critically depends on the relative timing of pre- and postsynaptic spikes on a millisecond time scale [4, 5]. Accordingly, STDP time constants of the order of 10–40 ms were widely used in previous STDP modeling studies [1, 107, 108, 111, 112].

We then stimulated the two neurons periodically with period $T$ (or alternatively with frequency $\nu = 1/T$) but at different times with a time shift $\Delta t$ in order to find which values of $T$ and $\Delta t$ lead to a depression of both synapses. First, we ignored the delay in the transmission of the signals between the two neurons (i.e., when $|\xi| = 0$). Over a period each synapse experiences a succession of potentiation and depression and the net change of the synaptic strength depends on the superposition of the two changes [106, 113]. For more clarity we assumed that the neuron N1 in Fig 1A is stimulated first and the neuron N2 is stimulated after the time shift $\Delta t$. The $1 \rightarrow 2$ synapse experiences a potentiation ($\Delta g_{21}^+$) and a depression ($\Delta g_{21}^-$), and it will be depressed if $\Delta g_{21}^+ < \Delta g_{21}^-$. By the same token, the reverse synapse (i.e., $2 \rightarrow 1$) is depressed when $\Delta g_{12}^+ < \Delta g_{12}^-$. Taken together, the condition for depression of synapses in both directions will be given by $\Delta g_{21}^{(T)}(\Delta t) < 0$ and $\Delta g_{12}^{(T)}(\Delta t) < 0$ (see Eq (5) in Methods).

The regions for depression (blue) and potentiation (red) of both synapses and the potentiation of one synapse and the depression of the other (orange) are shown Fig 2 in the $\Delta t$-$T$ plane, as predicted theoretically by calculating the net synaptic change for reciprocal synapses between two neurons given by Eq (5). The results indicate that for the chosen STDP parameters, for most values of $\Delta t$ and $T$ the synapses will be potentiated in one direction and will be depressed in the other, leading to unidirectional connectivity. For the simultaneous depression of synapses in two directions, which is the target of our time-shifted stimulation approach, there is a desired range for $\Delta t$ and $T$ where the neurons should fire almost in anti-phase, i.e., $\Delta t \approx T/2 = 1/2\nu$. The numerical results shown in S3 Fig for an exemplary set of parameters (marked by point b in Fig 2A) verify the theoretical prediction, valid in the absence of axonal and dendritic delays.

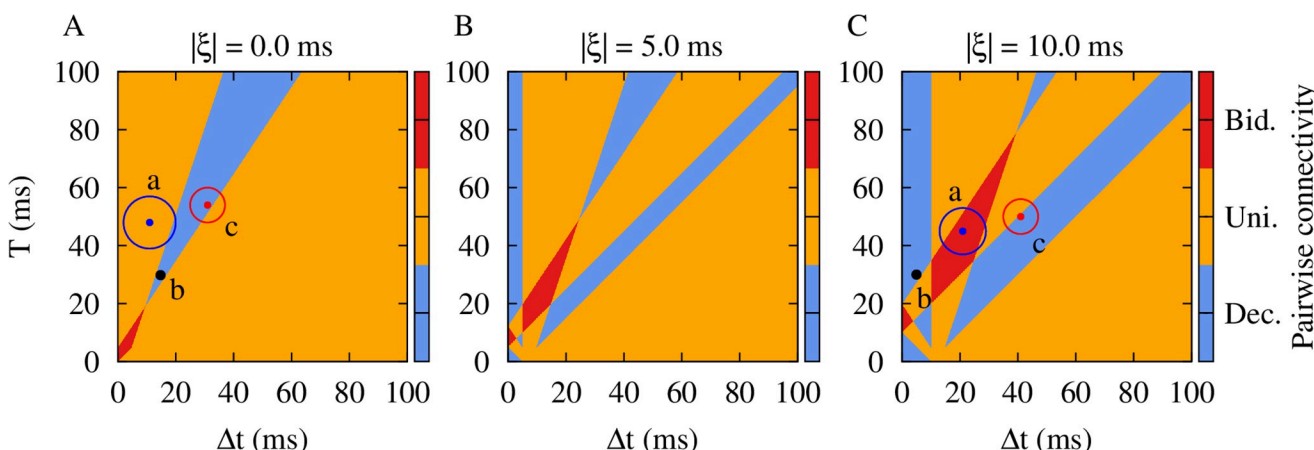

**Fig 2. Theoretical prediction of the emergent two-neuron connectivity.** Qualitative colors show the synaptic structure between a pair of pre- and postsynaptic neurons calculated based on the synaptic change in Eq (5) over a period ($T$): Decoupled (blue), unidirectional (orange) and bidirectional (red) regimes. STDP parameters were $A_+ = 0.008$, $A_- = 0.005$, $\tau_+ = 10$ ms and $\tau_- = 20$ ms. (**A**) Points a: (11, 48) ± 9 ms, b: (15, 30) ms and c: (31, 54) ± 6 ms represent ($\Delta t$, $T$) pairs obtained from simulations performed in S1 Fig before, during and after the stimulation, respectively, for $|\xi| = 0.0$ ms. (**B**) $|\xi| = 5.0$ ms; the effective delay at synapse reshapes the $\Delta t$-$T$ plane. (**C**) Points a: (21, 45) ± 8 ms, b: (5, 30) ms and c: (41, 50) ± 6 ms show the same ($\Delta t$, $T$) pairs as in A, but obtained from numerical simulations performed in Fig 3 before, during and after the stimulation, respectively, for $|\xi| = 10.0$ ms.

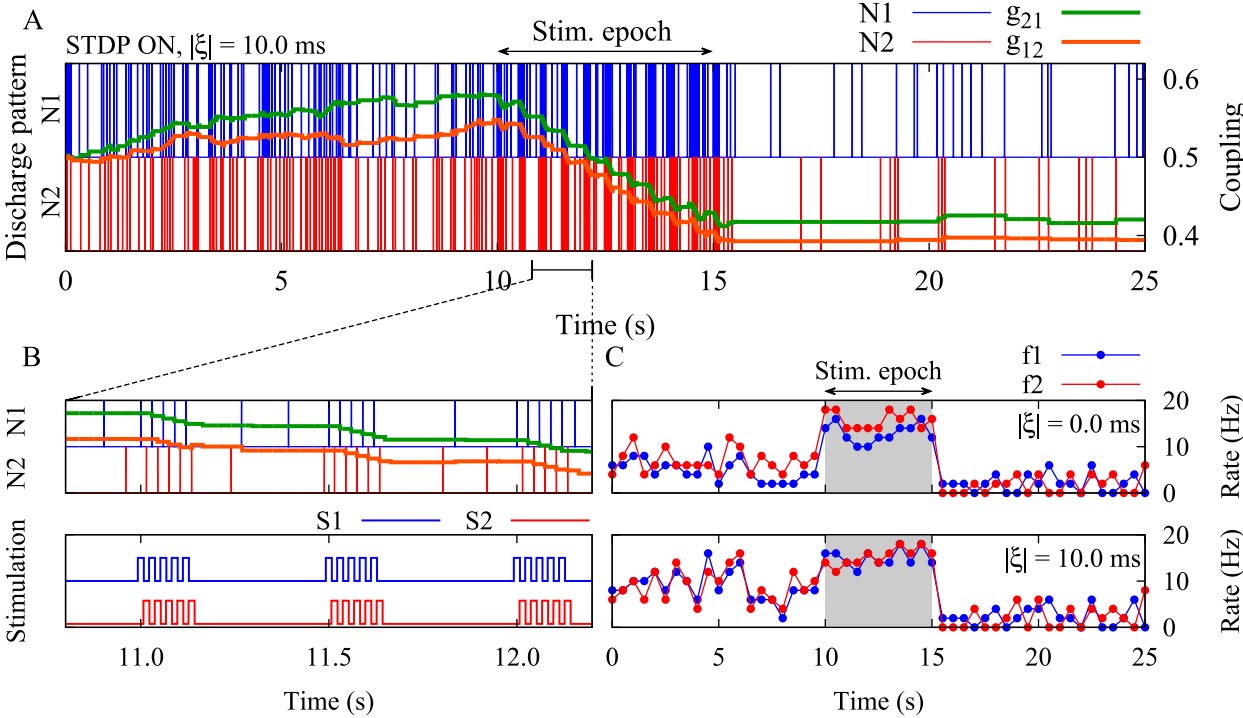

**Fig 3. Suppression of the synaptic strengths between two neurons by time-shifted stimulation.** (**A**) Time course of neuron discharges (N1/N2) and the synaptic strengths ($g_{21}/g_{12}$) are shown for two neurons when STDP parameters were $A_+ = 0.008$, $A_- = 0.005$, $\tau_+ = 10$ ms and $\tau_- = 20$ ms, and $|\xi| = 10.0$ ms. (**B**) (Top) Results in panel A are shown with a higher resolution. (Bottom) Stimulation signals were delivered to N1 (S1) and N2 (S2) for the duration of $T_{stim} = 5$ s (stimulation ON period in A) with time shift $\Delta t = 5$ ms and frequency $\nu = 33.3$ Hz (inspired by parameters shown in point b in Fig 2C). (**C**) Time course of the mean firing rate of neurons ($f1/f2$) for $|\xi| = 0.0$ ms (top) with ($\Delta t$, $T$) = (15, 30) ms and for $|\xi| = 10.0$ ms (bottom) with ($\Delta t$, $T$) = (5, 30) ms.

Next, we explicitly considered the role of realistic transmission delays in the model. As shown previously [106, 113], the difference between the dendritic and axonal delays ($\xi = \tau_d - \tau_a$), enters the formula for the modification of the synapses in Eq (4), since the effect of the firing of pre- and postsynaptic neurons builds up at the synapse after axonal and dendritic (back-propagation) transmission delays, respectively. In this case, the condition for depression of the synapses in two directions is given by $\Delta g_{21}^{(T)}(\Delta t + \xi) < 0$ and $\Delta g_{12}^{(T)}(\Delta t + \xi) < 0$. In Fig 2B and 2C, regions for three different kinds of final pairwise synaptic connections are shown for two different values of $|\xi|$, representing median transmission delay between cortical populations [114]. We considered positive values of $\xi$ since for long-range connections axonal delays are greater than dendritic delays. Intriguingly, at small values of the firing period (i.e., at high stimulation frequencies), and also at greater periods (i.e., at low stimulation frequencies), a close-to in-phase firing may lead to depression of both synapses. This is illustrated by a numerical experiment in Fig 3A and 3B with exemplary parameters marked by point b: The simulation related to ($\Delta t$, $T$) = (5, 30) ms in Fig 2C demonstrates the validity of the analytical predictions. Note, with this set of parameters coincidence of the spontaneous firing of the neurons led to a potentiation of both synapses (point a in Fig 2C). In contrast, stimulation with appropriate parameters (point b in Fig 2C) led to a depression of both synapses after stimulation offset (point c in Fig 2C). Obviously, administration of additional trains of stimuli leads to a further depression of the synapses. Fig 3C also shows that in the two-neuron motif either with (bottom) or without (top) transmission delay, the time-shifted stimulation effectively reduced the mean firing rates in both cases. This lead to a desynchronization of the two-neuron motif by

reducing the coincidence of the neuronal discharges where the neurons were unable to resynchronize their activity due to the weakened coupling.

## Bidirectionally connected populations

We then replaced each neuron in the two-neuron motif by a population of excitatory and inhibitory neurons and studied whether the theoretical predictions obtained in the two-neuron motif were valid for two bidirectionally connected populations (i.e., two modules in Fig 1C). The approach employing two-neuron motifs and then translating the results to two interacting populations was successfully used previously in several computational studies [108, 115, 116]. By the same token, several studies used neuronal network models comprising two populations interacting via long-range excitatory-to-excitatory delayed projections modified by STDP [84, 100]. Weakly coupled excitatory and inhibitory neurons within each population were connected in a sparse manner. The synaptic strengths, transmission delays and external input were tuned such that each population operated in an inhibition-stabilized regime, characterized by irregular individual firing of neurons (see S1(B) Fig) [84, 102, 104]. However, the network operating point was close enough to the oscillatory regime such that strong long-range excitation could elicit pathological synchronous firing [18, 117] of the two modules with high firing rate [84, 86, 99], as consistent with anatomical evidence that cortical excitation has a larger spread than local inhibition due to long-range projections of cortical pyramidal neurons [118, 119]. As shown previously, when this long-range excitation is sufficiently strong, the network dynamics exhibit synchronous oscillations, whereas decoupling of the two populations deteriorates collective oscillations [86, 99]. Stimuli were delivered to the two populations separately, thereby homogeneously affecting all excitatory and inhibitory neurons in each population, respectively. By stimulating both populations with time-shifted stimulus pulse trains, we aimed at inter-population decoupling, i.e., reduction of the strong, plastic synaptic connections between the two populations, in this way inducing an effect outlasting the cessation of stimulation. This, in turn, caused a desynchronization of the populations. The parameters of the stimulation, including frequency and time shift, were chosen based on our results obtained in a pair of neurons as shown in Fig 2.

We first examined the system in the absence of any transmission delay. In that case, the stimulation can suppress the inter-population connections and desynchronize the neuronal activity in the populations in a long-lasting manner, exceeding cessation of stimulation (see S4 Fig). Note, taking into account realistic transmission delays changes the stimulation parameters required for decoupling and desynchronizing the two cortical modules (see Fig 2C). Experimental estimates of delays in long-range inter-population connections vary from a few to tens of milliseconds [120] and, hence, cannot be ignored in biologically realistic simulations. As shown previously [112, 121], in the absence of transmission delays synapses tend to evolve in an asymmetric manner, giving rise to unidirectional connections, where depression of a synapse usually comes at the expense of potentiation of the reverse synapse. Therefore, bidirectional depression of synaptic strengths is unlikely to occur in the absence of transmission delays, especially for balanced STDP profiles [112, 121]. In contrast, in the presence of transmission delays, simultaneous bidirectional depression or potentiation of reciprocal synapses is more likely to occur (see Refs. [106, 113, 122] and Fig 2A-C).

Inter-population delays represent the delay in the transmission of signals between the two distant brain regions which interact through long-range projections. Such a range of transmission delays were experimentally observed in cortico-cortical connections, e.g., between primary somatosensory cortex (S1) and secondary somatosensory cortex (S2) in rabbits ($\sim$ 2–30 ms) [123] and cats ($\sim$ 2–40 ms) [124]. In humans, the average-sized myelinated fiber

interconnecting the temporal lobes would have an inter-hemispheric delay of over 25 ms [125, 126]. The inter-module delays in our study are chosen to lie within this range of realistic delays for long range connections, but the choices are exemplary (e.g., 5 and 10 ms) and the results can be easily adopted for two specific areas with a known transmission delay.

In this study, we fixed the dendritic (back-propagation) delay at a biologically realistic value $\tau_d$ = 0.5 ms, whereas the remaining delays were assigned to the axonal delay. The dynamical and structural characteristics of the network before and after stimulation are shown in Fig 4. The initial values of the connection strengths were chosen based on the previous results [113]. The neurons in the two populations fired in an irregular and asynchronous manner when the two populations were isolated, while they fired synchronously and the populations oscillated in an anti-phase manner when the two populations were connected by the long-range excitatory projections (Fig 4A1). Other phase relationships could be obtained depending on the choice of the inter-populations delays (see S4 Fig). This models of a pathological state with strong phase-locked oscillations of the two populations (Fig 4D, left), regular spiking (Fig 4C, grey) at a high rate (Fig 4H) and high pairwise correlation (Fig 4B, grey) between the spiking activities of the neurons in each module. Based on the values of the periods and the time-shift between the stimulus trains in the two-neuron motif (Fig 2), the outcome of stimulation can be predicted. The simulation results for the two populations (Fig 4F and 4G) are in accordance with the prediction of the theoretical results (Fig 2C). To induce an unlearning of the pathological state, we choose stimulation parameters from the two-neuron motif given by point b in Fig 2C in order to decouple the two populations (frequency $v$ = 1/$T$ = 33.3 Hz, time shift $\Delta t$ = 5 ms and delay $|\xi|$ = 10.0 ms). After 5 s of stimulation, the mean synaptic strengths between both cortical modules were significantly suppressed in both directions (Fig 4F). Accordingly, right after stimulation the initially pronounced oscillations of both cortical modules are suppressed (Fig 4A2 and 4E, colored), and the neurons fire at a much lower rate and in an irregular manner (Fig 4B, 4C and 4H, colored).

The persistent effect of the stimulation on desynchronization and decoupling might be of great clinical interest. It appears the persistent effect was driven by lower firing rates and uncorrelated firing of the neurons after the cessation of stimulation. In this case, it is important to know under what conditions the firing rate and the synchrony remains low after stimulation ceases. The noise level [84] and strength of the local excitatory and inhibitory connections [102, 104] may actually play a role in this process; however, we intended to highlight the role of inter-population connections in the emergence and disappearance of the pathologically synchronous activity. To this end, the intra-population synaptic strengths, transmission delays and external input were tuned such that each isolated population worked in a normal, irregular and asynchronous condition where the initially strong synaptic connections between the two populations led to the synchronous firing of the two modules with high firing rate. Our observations in Fig 5 revealed that stimulation-induced reduction of the inter-population synaptic strengths below a threshold prevents the re-potentiation of synaptic strengths and, thereby the re-synchronization of the population activity where the firing rates remain low after the stimulation cessation. This results in the low-frequency and uncorrelated firing of neurons within each population following stimulation which leads to a depression-dominated modification of the synaptic strengths (according to the STDP profile) and, thereby stabilizes the network's dynamics and connectivity in a loosely connected state.

This realizes the decoupling and desynchronizing effects of the stimulation. In fact, stimulation of both populations with a time shift results in a significant reduction of inter-population coupling strength through STDP. This causes a desynchronization of the populations due to a decay of the external drive from the reciprocal population [127]. In this way, the decoupled populations return to their desynchronized firing activity in the absence of a significant

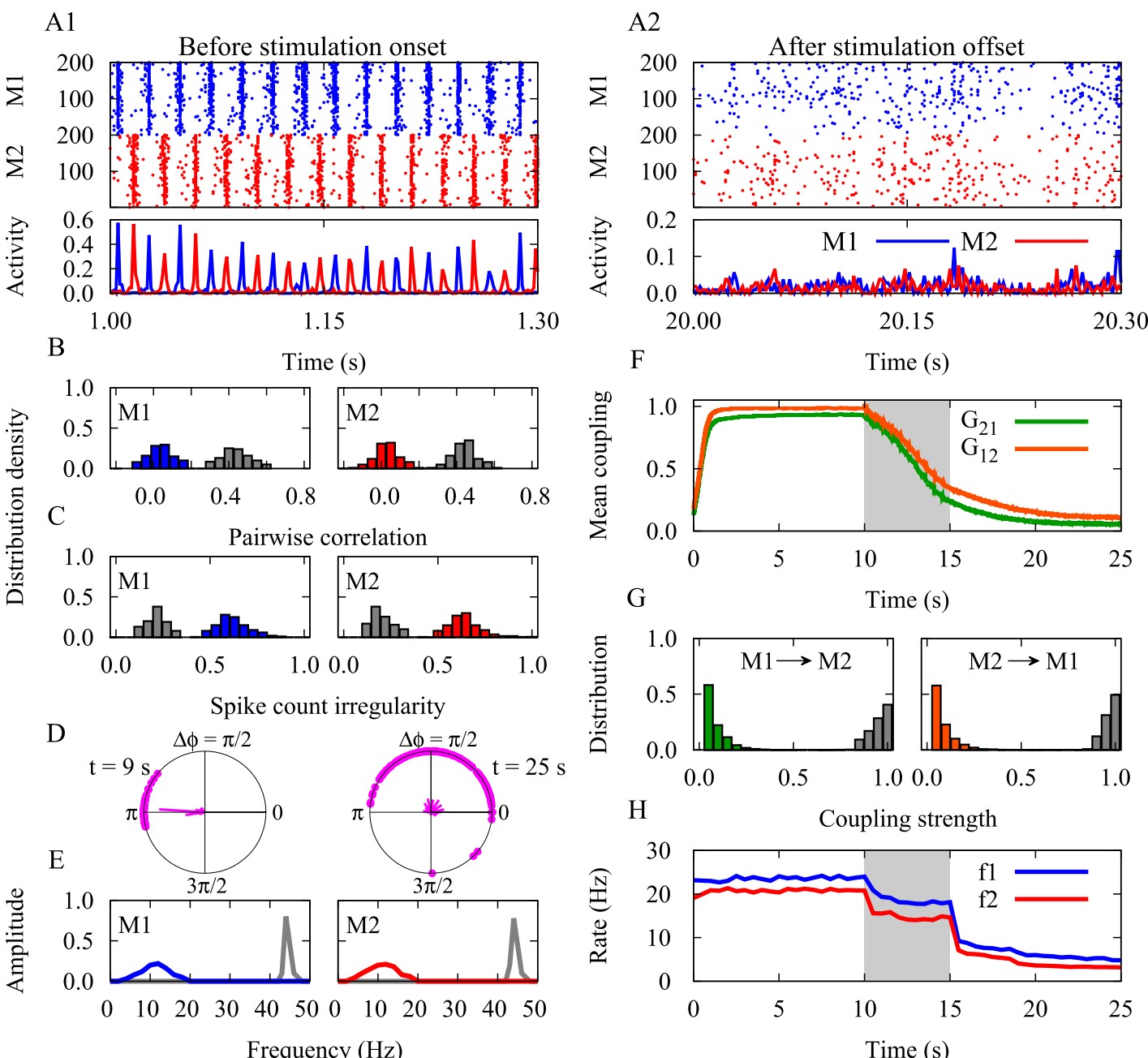

**Fig 4. Decoupling by time-shifted stimulation in the network.** (**A1,A2**) Raster plots and population activities (percentage of neurons firing per time window) are shown for the modules (M1/M2) before/after stimulation on/offset. (**B,C**) Distribution of the pairwise correlation and spike count irregularity of each module before (grey) and after (colored) stimulation epoch. (**D**) Snapshot of phase lags ($\Delta\phi$) between synchronous discharges from the two modules before $t = 9$ s, left) and after ($t = 25$ s, right) stimulation epoch. The radial bars show the distribution of the phase lags. (**E**) Fourier transform frequency of the population activity of each module before (grey) and after (colored) stimulation epoch. (**F**) Time course of the inter-population mean coupling ($G_{21}/G_{12}$). (**G**) Distribution of the inter-population synaptic strengths before (grey) and after (colored) stimulation epoch. (**H**) Time course of the mean firing rates ($f1/f2$) of neurons in each module. The modules were stimulated with time shift $\Delta t = 5$ ms and frequency $\nu = 33.3$ Hz (point b in Fig 2C) for the duration of $T_{stim} = 5$ s (highlighted area in F and H). The effective delay was $|\xi| = 10.0$ ms. STDP parameters were $A_+ = 0.008$, $A_- = 0.005$, $\tau_+ = 10$ ms and $\tau_- = 20$ ms.

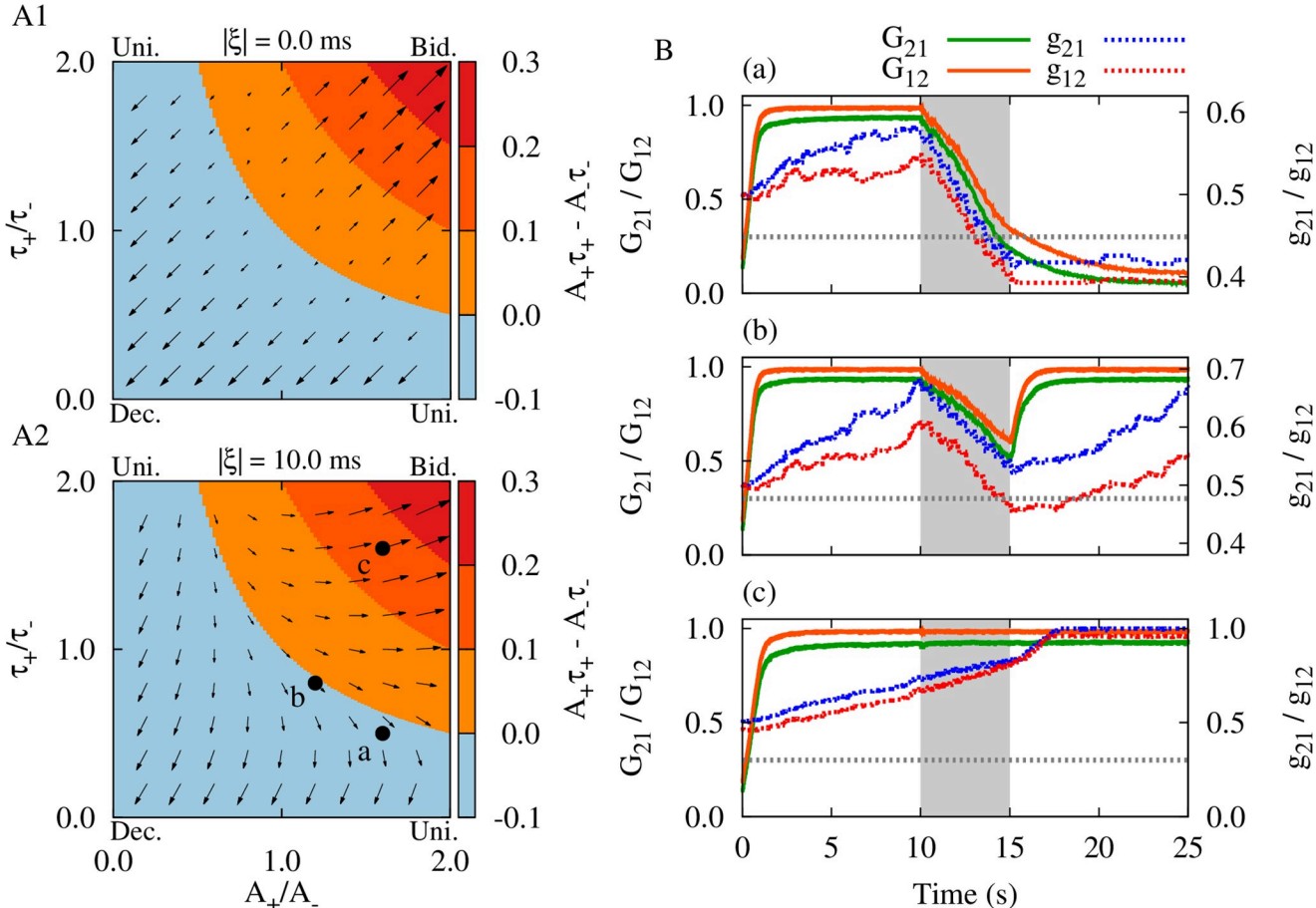

**Fig 5. Stimulation-induced decoupling depends on the STDP parameters. (A1,A2)** Colors show the imbalance between STDP potentiation and depression rates, and time constants for $|\xi|$ = 0.0 ms (A1) and $|\xi|$ = 10.0 ms (A2). $A_-$ = 0.005 and $\tau_-$ = 20 ms were fixed and $A_+$ and $\tau_+$ were varied. Arrows show the direction of synaptic change calculated theoretically based on Eq (5) where each corner of A1 and A2 labeled as the emergent synaptic structure between two neurons marked as bidirectional (right top), unidirectional (left top and right bottom) and decoupled (left bottom) states. Point a: $(A_+/A_-, \tau_+/\tau_-) = (1.6, 0.5)$ shows parameters used for stimulation in Figs 3 and 4. Parameters shown in points b: (1.2, 0.8) and c: (1.6, 1.6) were used to provide a comparison. **(B)** Time course of the synaptic strengths in the two-neuron motif $(g_{21}/g_{12})$ and inter-population mean coupling between two modules in the network $(G_{21}/G_{12})$ are shown for STDP parameters represented by points a-c in A2, respectively. The highlighted area indicates the stimulation epoch. The horizontal dotted lines (grey) roughly show the threshold $(G_{th} \approx 0.3)$ where the synaptic strengths must cross to obtain a stable loosely connected network structure. Stimulation parameters were $\nu$ = 33.3 Hz and $\Delta t$ = 5 ms.

external drive where the individual neurons fired stochastically. These irregular activity states reflect basic properties of normal cortical states [105, 128, 129].

Moreover, the initial distribution of the synaptic weights was chosen to give rise to pathological dynamics. However, as it has been shown previously, that the mean of the initial distribution of the synaptic weights or its standard deviation could, in principle, affect the emerging dynamics and structure due to multistability of synchronized and desynchronized states [130] or multistability of weak and strong connectivity regimes in plastic neuronal networks [113].

## Long-lasting decoupling by STDP

Intriguingly, after stimulation offset the connections continue to weaken. This is a very important point for the design of stimulation protocols that enable long-lasting effects. To avoid a relapse of the pathological dynamics the system has to be shifted into the basin of attraction of

the desynchronized state, rather than still remaining in regimes supporting repotentiation of the synapses (i.e., red or orange zone in Fig 2). In fact, the targets of our stimulus protocol are the basins of attraction of favorable, desynchronized states (cf. Ref. [131]) with STDP profiles satisfying the condition $A_+\tau_+ < A_-\tau_-$ [107] (blue region in Fig 5A1 and 5A2), where the irregular neuronal firing and the low inter-module correlation may enable a continued depression of the synapses between both cortical modules. We hypothesized that the decrease of the synaptic weights below a certain threshold can prevent from re-potentiation of the synapses and the corresponding relapse of pathological dynamics after the cessation of stimulation.

To test this hypothesis and to demonstrate the predictive ability of the two-neuron results we repeated the simulations for both the two-neuron motif and the bidirectionally connected neuronal populations with different sets of STDP parameters (three sets are shown in Fig 5A2). Changing the ratio of potentiation and depression rates and time constants leads to different trade-offs between potentiation and depression regimes. For illustration, we used the same stimulation protocol with fixed parameters ($\nu$ = 33.3 Hz and $\Delta t$ = 5 ms) and a given delay $|\xi|$ = 10 ms, and observed qualitatively different outcomes by changing the STDP parameters: Simultaneous depression of the synapses (Fig 5Ba and 5Bb) or simultaneous potentiation of the synapses (Fig 5BC). While $A_+\tau_+ < A_-\tau_-$ was ensured in Fig 5Ba and 5Bb, the mean rates of the depression of the synapses were different because of the different STDP parameters.

Whether or not the strong synaptic connections between the two neurons or two populations are weakened during stimulation and continue to weaken after the cessation of stimulation depends on the STDP parameters. The depression of the synaptic strengths below a threshold leads to a continued weakening of synaptic connections outlasting stimulation offset (Fig 5Ba). By changing the STDP parameters, it might take longer to enter the targeted basin of attraction. For instance, if at stimulation offset not all synaptic weights were below threshold, the connections got re-potentiated thereafter (Fig 5Bb). In contrast, in Fig 5BC the stimulation was unsuccessful since for the selected set of STDP parameters the stimulation induced a potentiation of the synapses. These behaviors were fairly predicted by the direction of the synaptic change (indicated by arrows) in Fig 5A1 and 5A2 calculated from Eq (5) for the two-neuron motif where each corner in panels A1 and A2 marks the emergent pairwise synaptic structure. A successful, long-lasting decoupling stimulation requires the stimulation frequency and time shift to be chosen in a way that STDP depression parameters dominate over potentiation. In addition, a sufficient amount of stimulation has to be delivered to ensure that the synaptic weights fall below threshold in order to cause a desynchronization of the populations.

The final network topology after the cessation of stimulation can be roughly estimated based on the theoretical results in Fig 2 and simulation results in Fig 5, depending on the STDP parameters. The initial structural connectivity matrix of the network is shown in S5A1 Fig, using a binary representation which follows a random degree distribution (S5A2 Fig). To find the final network topology based on the synaptic strengths (S5B1 Fig), we transformed the structural connectivity matrix into a binary connectivity matrix shown in S5B2 Fig, by introducing a threshold over which the link is maintained in the binary matrix. The final synaptic strength matrix and the final connectivity matrix of the network are shown in S5B1, S5B2, S5C1 and S5C2 Fig for two different sets of the STDP parameters used in Fig 5Ba and 5BC, respectively. Particularly, successful stimulation-induced decoupling results in a significantly reduced connectivity (S5B2 Fig) where most of the long-range connections are depressed (S5B1 Fig) so that they are eliminated in the binary connectivity matrix (S5B3 Fig). In contrast, unsuccessful stimulation-induced decoupling is associated with significantly increased synaptic strengths (S5C1 Fig) where the final connectivity is equivalent to the initial structural connectivity (cf. S5C2 and S5A1 Fig), both following the same random degree distribution (cf. S5C3 and S5A2 Fig).

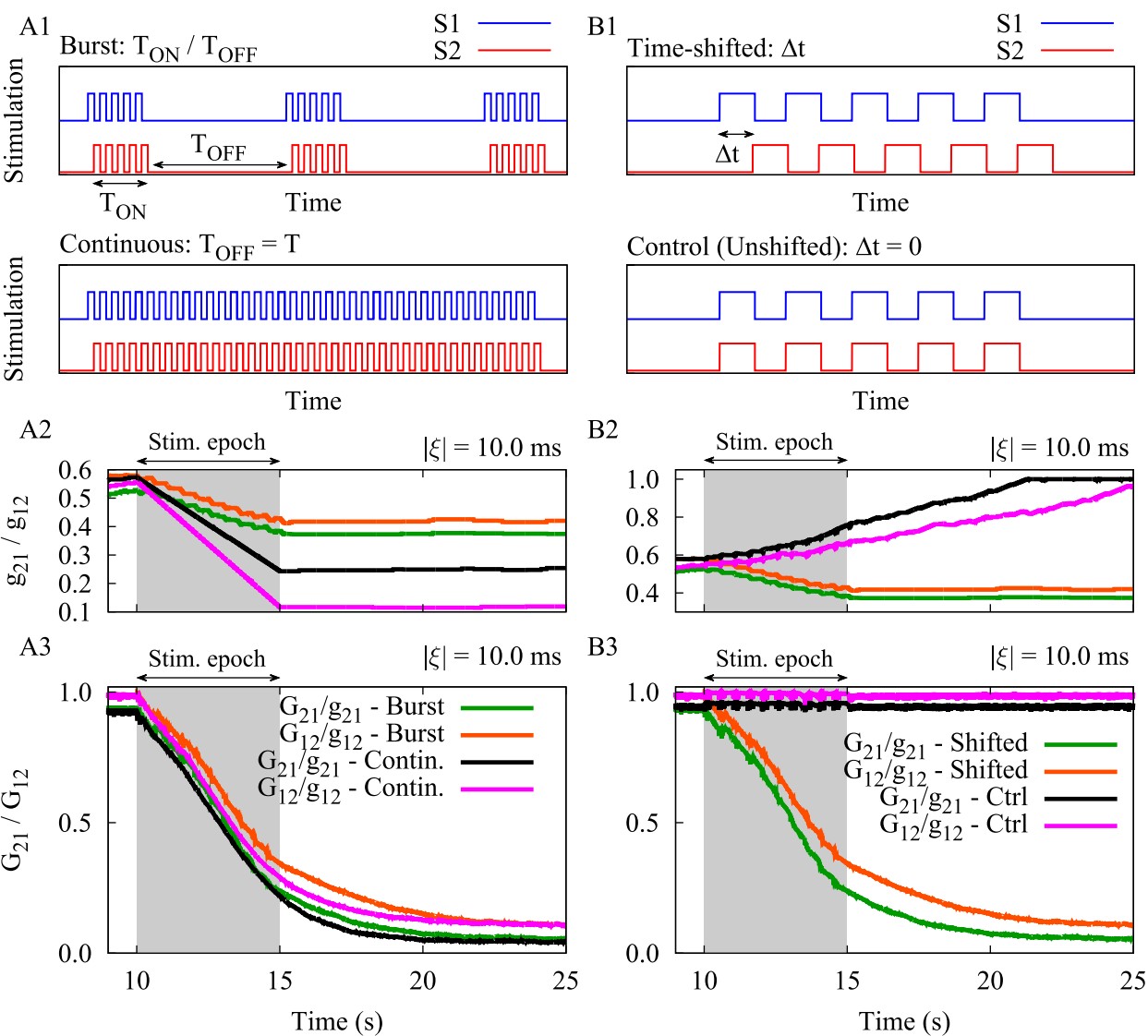

**Fig 6. The effect of stimulation pattern on the stimulation outcome.** (**A1**) Burst stimulation pattern (top) vs. continuous stimulation pattern (bottom) with the same frequency and time shift between the two signals (i.e., $\nu = 33.3$ Hz and $\Delta t = 5$ ms). (**A2,A3**) Time course of the synaptic strengths in the two-neuron motif ($g_{21}/g_{12}$) and the inter-population mean coupling between two modules in the network ($G_{21}/G_{12}$) are shown for burst and continuous patterns delivered to each module for $|\xi| = 10.0$ ms. (**B1**) Time-shifted stimulation (top) vs. control coincident stimulation (bottom) with the same frequency ($\nu = 33.3$ Hz). (**B2,B3**) Same as A2 and A3, but for the burst stimuli delivered with and without time shift. The highlighted area indicates the stimulation epoch. STDP parameters were $A_+ = 0.008$, $A_- = 0.005$, $\tau_+ = 10$ ms and $\tau_- = 20$ ms.

## Burst stimulation vs. continuous stimulation

So far, we used a time-shifted burst stimulation pattern (see Fig 6A1, top) in order to suppress the synaptic connections between the two neurons (Fig 3A) or the two modules (Fig 4F). In this case, the depression of the synaptic strengths at the end of each stimulation epoch occurs in a step-like manner due to the ON/OFF epochs of the burst stimulation ($T_{ON}/T_{OFF}$ in Fig 6A1, top). When the stimulation parameters, i.e., time shift, frequency and integral amount of stimulation, are adequately tuned based on the given STDP parameters, stimulation of sufficient duration can effectively decouple strongly connected neurons in the two-neuron motif as well as the two-module network.

To demonstrate whether mutually shifting the stimulation pulse patterns in time by $\Delta t$ may work in a rather generic way, we additionally considered a continuous stimulation pattern (Fig 6A1, bottom), implemented by eliminating the OFF-epoch ($T_{\text{OFF}}$) from the stimulation protocol by setting $T_{\text{OFF}} = T$ in Eq (3). Note that the continuous stimulation was applied to the system with the same intra-burst frequency (i.e., $\nu = 33.3$ Hz) as of the burst stimulation. In both scenarios, the stimulation signals delivered to the two neurons or two modules were time-shifted by $\Delta t = 5$ ms. As one can hypothesize, the application of a continuous stimulation pattern in this setting expedited the rate of decoupling due to the elimination of $T_{\text{OFF}}$ (a time window enabling the synaptic strengths to re-increase) as it is shown in Fig 6A2 and 6A3. In reality, however, the continuous stimulation might invoke slower secondary processes such as adaptation which might counteract the desired effects. However, this remains to be tested experimentally. On the other hand, the integral current delivery is much higher in the case of continuous stimulation. The results obtained with continuous stimulation indicate that the synaptic weight reduction induced by time-shifted stimulus delivery may be a mechanism that applies to a larger class of stimulation patterns, not only to periodic delivery of bursts.

## Time-shifted stimulation vs. unshifted stimulation

We illustrated that by introducing a simple time shift between the stimulation signals applied to two neurons or to two bidirectionally connected neuronal populations effectively reduced strong synaptic connections between the populations and caused a desynchronization of the populations. In particular, to demonstrate the significance of the time shift in the burst stimulation pattern (shown in Fig 6B1, top) we considered a control condition, where bursts were delivered coincidentally, without time shift (shown in Fig 6B1, bottom) and studied its impact on the two-neuron motif and the two-module network.

We repeated the simulations for the two-neuron motif and the two-module network, thereby taking into account realistic transmission delays in the model. The control condition was implemented by setting $\Delta t = 0$. In this case, spontaneous background activity may cause jitters between in-phase stimulation pulses resulting in the emergence of small time lags between neuronal spikes which can lead to the potentiation of the synapses in both directions since $A_+ > A_-$ in Eq (5). Therefore, in this setting the type of synaptic modification is determined by the dominance of the STDP-induced potentiation over STDP-induced depression rather than the STDP time constants ($\tau_+ < \tau_-$) that are important at greater time lags. As shown in Fig 6B2 and 6B3, the control stimulation induced a reciprocal potentiation of the synapses between the two neurons in the two-neuron motif and, by the same token, potentiated the inter-population synaptic connections between the two bidirectionally connected neuronal populations (cf. green/orange curve with black/magenta one in Fig 6B2 and 6B3).

Note that the emergent connectivity patterns shown in Fig 2 are theoretical predictions (Eq (5) in Methods) based on deterministic single neuron dynamics evaluated over the time interval between two regular spikes [106]. While the theoretical results roughly predict the stimulation-induced emergent structure of the two-neuron motif and the two populations, the irregular firing of the neurons in the case of two populations makes deviations from the theoretical results. In fact, during stimulation most of the neurons are forced to spike at the stimulation frequency so that the theoretical predictions are supposed to hold. However, at the extreme points, e.g., at $\Delta t = 0$ as in Fig 6B2 and 6B3, the synaptic strengths change only due to the jittering around $\Delta t = 0$ where the potentiation dominates since $A_+ > A_-$. Note that such an effect of jittering which is not reflected in Fig 2, is only significant for $\Delta t = 0$ and in general for the points lying in the borders of the potentiation/depression areas.

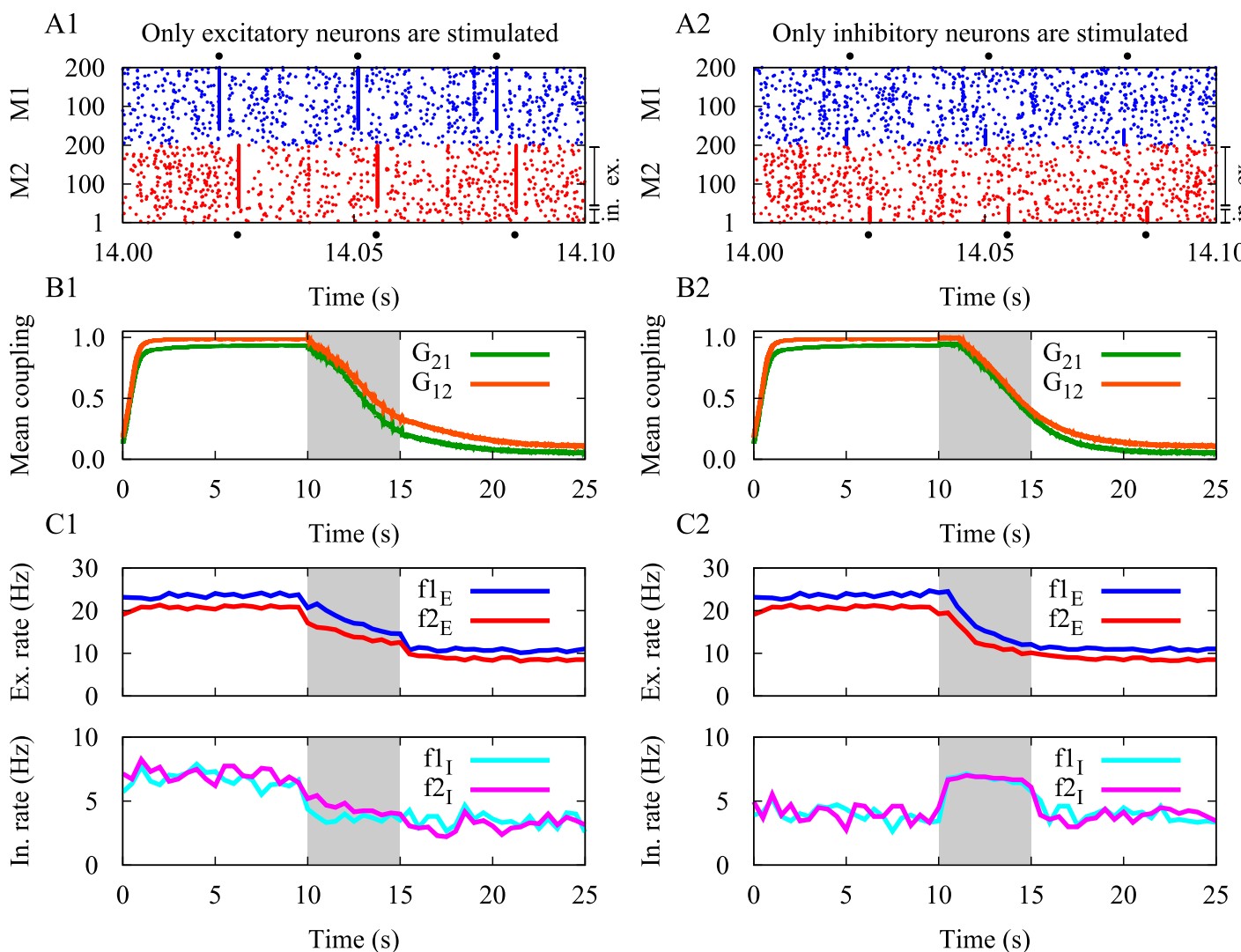

**Fig 7. Targeting excitatory cells or inhibitory cells during stimulation.** Stimulation outcome when only excitatory neurons (left column) and only inhibitory neurons (right column) were stimulated. (**A1,A2**) Raster plots are shown for the modules during a 100 ms window of the 5 s stimulation epoch. In each module, neurons #1 to #40 are inhibitory and the remaining are excitatory. For clarity, the onset times of stimulation are indicated with black circles around each panel. (**B1, B2**) Time course of the inter-population mean coupling. (**C1,C2**) Time course of the mean firing rates of excitatory (top) and inhibitory (bottom) neurons in each module. Stimulation and STDP parameters were similar to those used in Fig 4 (see Tables 2 and 3).

## Targeting excitatory cells or inhibitory cells during stimulation

In our model, we assumed the external stimulation as a simple input current separately delivered to the two modules, affecting all excitatory and inhibitory neurons in each population, respectively. To investigate whether the stimulation outcome depends on the type of stimulated cells, i.e., excitatory cells vs. inhibitory cells, we specifically stimulated excitatory neurons in Fig 7A1 and inhibitory neurons in Fig 7A2 with the same stimulation and STDP parameters used in Fig 4 (see Tables 2 and 3). Slightly varied stimulation/model parameters led to a similar outcome (not shown). Interestingly, in either case the stimulation outcome is similar to the case where all excitatory and inhibitory neurons were stimulated (Fig 4). Particularly, the mean synaptic strengths between the two populations were significantly reduced after the cessation of stimulation (Fig 7B1 and 7B2). After 5 s of stimulation, both excitatory (Fig 7C1, top)

**Table 3. Summary of the stimulation model parameters.**

| Effective delay | Parameter | Symbol | Value | Figures |
|---|---|---|---|---|
| $\|\xi\| = 0.0$ ms | Stimulation time shift | $\Delta t$ | 15 ms | |
| | Unshifted stimulation | | 0 ms | |
| | Inter-pulse interval | $T$ | 30 ms | |
| | Intra-burst frequency | $\nu$ | 33.3 Hz | Fig 3C, top |
| | Burst stimulation ON-epoch | $T_{\text{ON}}$ | 120 ms | Fig 8A1–8A4 |
| | Burst stimulation OFF-epoch | $T_{\text{OFF}}$ | 360 ms | S3 Fig |
| | Continuous stimulation OFF-epoch | | 30 ms | S4 Fig |
| | Total duration of stimulation | $T_{\text{stim}}$ | 5 s | |
| | Number of pulses within a burst | $k$ | 5 | |
| $\|\xi\| = 10.0$ ms[*] | Stimulation time shift | $\Delta t$ | 5 ms | |
| | Unshifted stimulation | | 0 ms | Fig 3A |
| | Inter-pulse interval | $T$ | 30 ms | Fig 3C, bottom |
| | Intra-burst frequency | $\nu$ | 33.3 Hz | Fig 4 |
| | Burst stimulation ON-epoch | $T_{\text{ON}}$ | 120 ms | Fig 6 |
| | Burst stimulation OFF-epoch | $T_{\text{OFF}}$ | 360 ms | Fig 7 |
| | Continuous stimulation OFF-epoch | | 30 ms | Fig 8B1–8B4 |
| | Total duration of stimulation | $T_{\text{stim}}$ | 5 s | S5 Fig |
| | Number of pulses within a burst | $k$ | 5 | |

[*]Effective delay is given by $\xi = \tau_d - \tau_a$, where the dendritic delay was fixed at $\tau_d = 0.5$ ms and the remaining delays were assigned to the axonal delay $\tau_a$ [106].

and inhibitory (Fig 7C1, bottom) neurons fired at a lower rate when only the excitatory neurons were stimulated. Following stimulation of only the inhibitory neurons, the firing rate of inhibitory neurons increased during stimulation (Fig 7C2, bottom), reducing the firing rate of excitatory neurons during and after stimulation (Fig 7C2, top).

In a realistic situation, however, depending on whether the external stimulation represents tDCS/tACS or a sensory stimulus, it may affect excitatory and inhibitory cells differentially [132–136]. For instance, inhibitory neurons can be activated at low stimulation intensities, whereas excitatory neurons would require stronger stimulation, as revealed by computational [132–134] as well as experimental findings [135, 136]. The stimulation frequency can also differentially modulate the firing rate of excitatory and inhibitory neurons. While at low frequency (i.e., 5 Hz), the firing rate of inhibitory neurons is preferentially increased, at higher frequencies (i.e., 26 and 52 Hz), the excitatory neurons show increased firing rates [133, 134], potentially due to the intrinsic difference in decay times of excitatory and inhibitory synapses [134]. Therefore, careful selection of specific frequencies and amplitudes of the stimulus may allow for selective enhancement and suppression of the excitation-inhibition ratio.

### Rescaling stimulation frequency and time shift

In order to study the influence of the stimulation time shift ($\Delta t$) and frequency ($\nu$) on the stimulation outcome we ran numerical simulations for a reasonable range of $\Delta t$ and $\nu$ and measured the averaged inter-population mean coupling between the two neuronal populations given by Eq (6) and the synchrony level by the pFF of the activity of each module given by Eq (10), as it is shown in Fig 8. The pFF evaluates the normalized amplitude of the variation of the population activity which increases when the neurons fire in synchrony [84, 110].

When transmission delays were not considered in the model (Fig 8A1–8A4), stimulation frequencies over 50 Hz with a small time shift (i.e., $\Delta t < 10$ ms) and stimulation frequencies

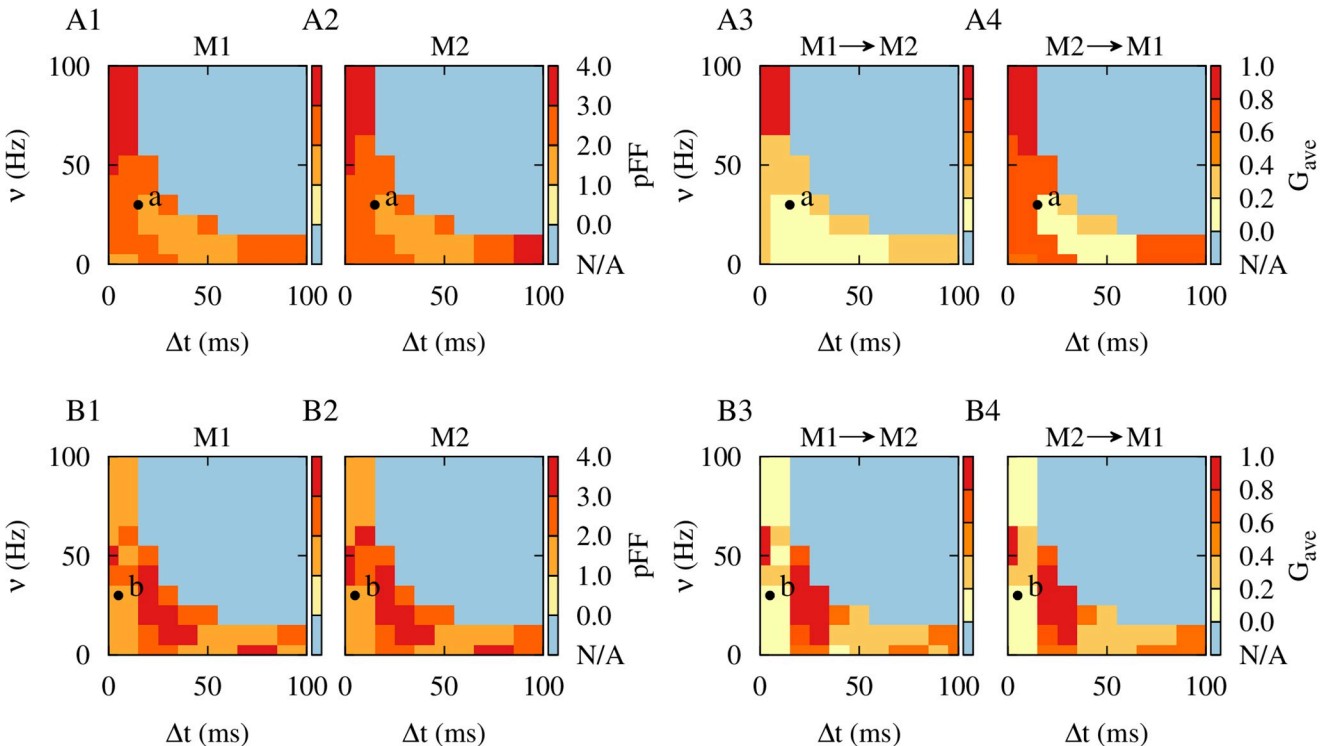

**Fig 8. Rescaling the parameters of time-shifted decoupling stimulation.** (**A1-A4**) Synchrony level measured by the pFF of the activity of each module (A1,A2) and the averaged inter-population mean coupling ($G_{ave}$) between the two modules in the network (A3,A4) obtained by numerical simulations for different values of stimulation frequency and time shift with $|\xi| = 0.0$ ms, evaluated over 10 s of activity after stimulation offset. (**B1-B4**) Same as A1-A4, but with $|\xi| = 10.0$ ms. STDP parameters were $A_+ = 0.008$, $A_- = 0.005$, $\tau_+ = 10$ ms and $\tau_- = 20$ ms. Points a and b in the figure indicate the stimulation frequency and time shift used in Fig 2A and 2C, respectively. Blue (N/A) region shows the range of parameters where the $0 < \Delta t < T = 1/\nu$ constraint is not satisfied.

below 20 Hz with a large time shift (i.e., $\Delta t > 70$ ms) failed to desynchronize neuronal activity in each module (Fig 8A1 and 8A2, dark red) or suppress strong inter-population connections between the modules (Fig 8A3 and 8A4, dark red). However, stimulation frequencies below 50 Hz with greater time shifts (i.e., 10 ms $< \Delta t <$ 70 ms) induced desynchronizing effects (Fig 8A1 and 8A2, light red) and suppressed inter-population connectivity in the network (Fig 8A3 and 8A4, light red). Point a characterized by ($\Delta t$, $\nu$) = (15 ms, 33.3 Hz) in Fig 8A1–8A4 shows the set of time shift and stimulation frequency used for decoupling stimulation in the absence of delays (see S3 and S4 Figs). The blue (N/A) region in Fig 8 shows the range of parameters where the constraint $0 < \Delta t < T = 1/\nu$ is not satisfied.

Inclusion of realistic transmission delays in the model crucially reshapes the optimal range of stimulation parameters required to induce decoupling effects (shown in Fig 8B1–8B4). In particular, in this case, stimulation frequencies over 60 Hz or below 30 Hz both with a small time shift (i.e., $\Delta t < 10$ ms = $|\xi|$) induced desynchronizing effects (Fig 8B1 and 8B2, light red) and suppressed inter-population connectivity (Fig 8B3 and 8B4, light red). This was also seen for stimulation frequencies below 20 Hz with medium range time shifts (i.e., 40 ms $< \Delta t <$ 80 ms). However, intermediate stimulation frequencies (i.e., 10 Hz $< \nu <$ 50 Hz) with time shifts in the range 20 ms $< \Delta t <$ 30 ms can be detrimental since they induce synchronized activity states with strong inter-population synaptic connections (Fig 8B1–8B4, dark red). Point b, given by ($\Delta t$, $\nu$) = (5 ms, 33.3 Hz), in Fig 8B1–8B4 corresponds to the time shift and stimulation frequency selected for decoupling stimulation in the presence of delays (see Figs 3 and 4).

The theoretically predicted multistability of decoupled (blue), unidirectional (orange) and bidirectional (red) connectivity regimes for the two-neuron motif in Fig 2 supports the stimulation-induced neuronal activity and synaptic connectivity emerging in the two-module network in Fig 8 fairly well. The inventory of these attractor states allows to choose the parameters of the time-shifted stimulation appropriately so that the stimulation pattern can lead to an unlearning of pathologically strong synaptic connectivity between the two modules and cause a desynchronization of the modules (light red in Fig 8), in this way inducing a sustained decoupling effect.

However slight deviations from these parameters produce undesirable effects so prediction precision will be important for translation. Summary of the stimulation model parameters is given in Table 3.

## Discussion

Our results illustrate that subtle changes of a stimulation protocol may have significant impact on the stimulation outcome. We presented a time-shifted two-channel stimulation approach in a generic cortex model that aims at decoupling two interacting neuronal populations by employing STDP, i.e., by unlearning pathologically strong synaptic interactions between the two populations. This simple intervention caused pronounced changes of the network dynamics and connectivity, outlasting the cessation of stimulation. In our model intra-population synaptic connections were weak and static and they had no role in the generation of synchronized oscillations and instead, synchronization of the populations was due to initially strong inter-population connections. Accordingly, decoupling of the two populations by the stimulation-induced reduction of inter-population excitatory projections below a threshold caused desynchronized and sparse activity of the neurons which is hallmark of the normal activity of cortical neurons [128, 129]. Remarkably, this state was stable and the pathological connectivity and dynamics did not relapse after cessation of the external stimulus.

In our model, each isolated module is an inhibition-stabilized local balanced network (see S1(A) Fig) [102, 103] and the two modules interact through long-range excitatory projections [137]. This fundamentally differs from a Hebbian-assembly model [103], i.e., a single excitatory population of neurons recurrently exciting itself. In our model, each local balanced network consists of both excitatory and inhibitory neurons in 4:1 proportion where the strength of inhibitory synapses within each module is on average 4-fold the strength of excitatory synapses. Connectivity and network parameters were tuned such that each population operated in an inhibition-stabilized regime, characterized by irregular individual firing of neurons and desynchronized network activity (see S1(B) Fig) [102, 104]. However, the network operating point was close enough to the oscillatory regime such that strong long-range excitation could elicit oscillatory global activity [84, 102, 104]. Accordingly, S1(C) Fig (top) demonstrates local balanced amplification (LBA) [103, 138] in an isolated module in our model which shows the firing rate of the excitatory ($r_E$) and inhibitory ($r_I$) populations in response to a pulse of input ($I_E$) to the excitatory population at time $t = 0s$ that sets $r_E = 10$ Hz ($r_I = 0$ Hz; representing the baseline of activity), similar to the protocol used by Murphy and Miller (2009) [103]. In this case, increase of the local recurrent excitation is stabilized with stronger inhibition through LBA so that excitation-inhibition ratio is finally balanced (also see S1(C) Fig, bottom) [103].

To reduce stimulation-induced side effects (see e.g. Ref. [52]) and to restore physiological function, it is desirable to reduce the integral amount and duration of stimulation. Accordingly, we do not only focus on acute effects (during stimulus delivery), but also on beneficial long-lasting effects, persisting after the cessation of stimulation [72, 73, 131]. Computationally, desynchronizing stimulation can shift adaptive networks from pathological attractor states

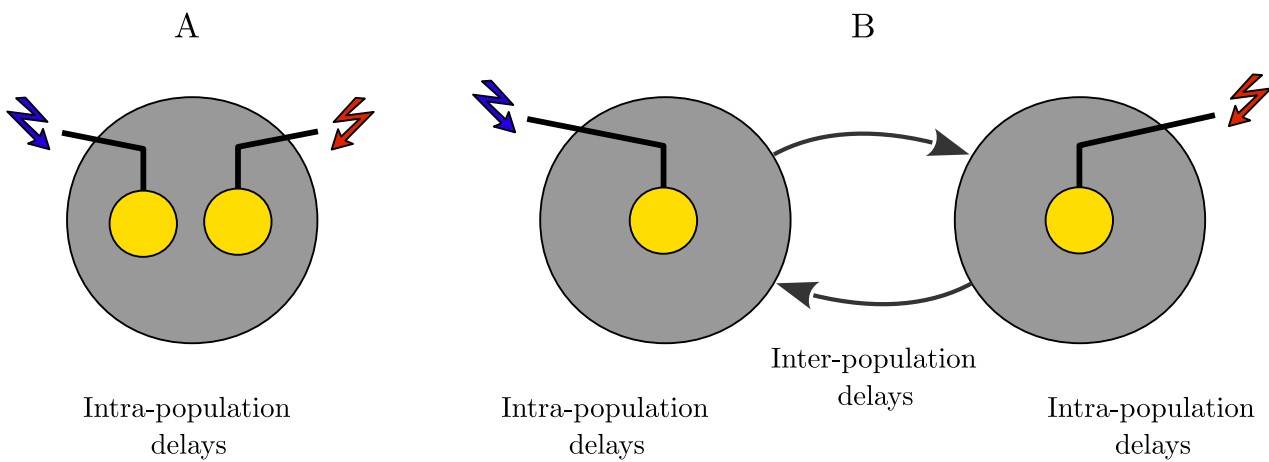

**Fig 9. Illustrative representation of the time-shifted stimulation.** (**A**) Schematically refers to a variety of bi-/multichannel stimulation techniques where the delays between different stimulation sites (yellow circles) are approximately the same. (**B**) Schematically refers to bi-/multichannel stimulation techniques with at least two qualitatively different delays, i.e., small delays within and long delays between stimulation sites.

towards physiological states [72, 80, 130]. In this way, therapeutic stimulation effects can be achieved that persist after discontinuation of stimulation [72, 80], as demonstrated in pre-clinical as well as clinical proof-of-concept studies [74–78]. However, the efficacy of desynchronizing stimulation crucially depends on the adaptation of stimulation parameters to the synchronization properties of the targeted network activity [80]. Furthermore, acute desynchronization does not necessarily lead to long-lasting changes of network activity [79, 80]. Apparently, enduring desynchronization can only be achieved if the pathological network connectivity is also modified by the external stimuli [79, 80], i.e., when the reduction of strongly synchronized activity is accompanied by a reduction of pathologically strong synaptic connectivity, to avoid relapse and reappearance of symptoms.

Control of synchronization in neuronal populations and rewiring of synaptic connections by stimulation was previously addressed in a number of computational studies. For example, different patterns of sequential activations of subpopulations of neurons with a small intra-population delay (time shift) between multiple stimulation sites (see Fig 9A) can induce an unlearning of pathological connectivity due to the desynchronization of neuronal activity in adaptive network models of phase oscillators [72] as well as a various neuron models [70, 130, 131, 139–141]. Another study focused on harnessing the underlying STDP in the network to reshape synaptic connectivity by applying dedicated stimuli in a two-population network model of excitatory and inhibitory LIF neurons [116]. Anti-phase delivery of charge-balanced stimulation pulses to the excitatory subpopulation and the inhibitory subpopulation with a small time shift led to a reduction of the average synaptic weight between the two subpopulations in the network along with a strong desynchronizeation [116]. More recently, it was shown that two-site stimulation of cortical populations with long inter-population delays (i.e. time shifts) between stimulation sites (see Fig 9B) can induce changes in the inter-population synaptic connectivity in a network of LIF neurons which are similar to the classical STDP profile [96]. Consistent with our results, the induction of significant plastic changes in inter-population connections was crucially affected by appropriate stimulus timing [96].

Cortical stimulation may offer simpler ways to modulate abnormally synchronized activity which is less invasive in comparison to DBS surgery. However, cortical stimulation may also have its limitations. For instance, electrical stimulation can intrinsically evoke

nonphysiological spatiotemporal patterns and complicated neural responses in cortical networks due to the transsynaptic activation of axons [95, 142]. Here, we used a simple and generic cortical model [81, 84, 98] to illustrate the critical role of the time shift in shaping long-lasting decoupling effects induced by stimulation. A simple model enables clear predictions based on a detailed investigation that is valid for a large parameter space, whereas complex and detailed computational models with many parameters may be difficult to study because of their high-dimensional phase space and the associated computational cost. Furthermore, we focused on the decoupling of the two populations achieved by the stimulation-induced reduction of the plastic synaptic connections between the populations and for simplicity assumed that the modules have only static synapses. However, assuming plastic synapses within each module may affect the presented results. In this case, the time-shifted stimulation should first decouple the two modules externally and, then, reduce the internal coupling of each module to ensure that long-lasting decoupling and desynchronizing effects can be achieved.

Introducing a time shift is a comparably simple procedure which does not require complex technical solutions, e.g., required for the detection of biomarkers, as in the case of demand-controlled stimulation [143]. By means of this maneuver (i.e., introducing the time shift), a variety of different bi-/multichannel stimulation protocols may have significantly different long-term effects. Our findings thus may have quite generic implications for different cortical stimulation strategies, e.g., through epicortical electrodes or intracortical electrodes. On the other hand, cortical networks may exhibit evoked or spontaneous collective oscillations in the absence of temporally structured stimuli, despite irregular and weakly correlated activity pattern of individual neurons, known as the normal state of the cortical activity [128, 129]. By the same token, physiological states in subcortical networks are characterized by uncorrelated neuronal activity as opposed to pathological states that are characterized by abnormally synchronized activity, for example, in PD [18]. This suggests that the results obtained for the simple cortical model may also be, in principle, applicable to basal ganglia (BG) models subjected to multichannel DBS protocols.

Interestingly, for the parameter ranges that we tested, qualitatively similar results were obtained for both the patterned delivery of bursts and trains of single stimulus pulses, implying that the time shift scheme works with burst stimulation as well as with continuous stimulation. This suggests that the reshaping of activity-connectivity patterns by time-shifted stimulus delivery may be a mechanism that applies to a larger class of stimulation patterns. This is encouraging for further test in more complex network models with more sophisticated intrinsic dynamics as well as in pre-clinical (animal) studies and clinical tests.

The role of the introduced time (phase) shift between stimulation signals delivered to the two populations is critical. Our results show that the time-shifted stimulation is more effective when the stimulation signals are almost in anti-phase (i.e., $\pi$ phase lag or $\Delta t \approx 1/2\nu$), whereas the almost in-phase stimulation (with zero phase lag or $\Delta t \approx 0$) failed to induce decoupling effects in our model. We used a typical STDP profile that is experimentally observed in cortex [7] and a reasonable range for cortico-cortical transmission delay [114], but in general, as predicted by our theoretical analysis, the effective time shift for stimulation deviates from anti-phase based on the inter-population transmission delay (see Fig 2) and the shape of the STDP learning window (see Fig 5). The phase-specificity of the decoupling effects in our model is consistent with previous experimental studies suggesting that in-phase cortical stimulation has a synchronizing effect on the neuronal activity in the target node, whereas anti-phase stimulation can cause desynchronizing effects [144]. Targeting multiple nodes of brain networks thus may enable stimulation strategies to improve stimulation outcome by phase manipulation between stimulation sites.

In this study, we focused on stimulation effects that persist after cessation of stimulation which are thought to be mediated by longer lasting models of synaptic plasticity such as STDP and, thereby, we neglected the potential role of short-term synaptic plasticity. However, stimulation-induced effects of the short-term synaptic plasticity can differentially modulate excitatory and inhibitory synapses depending on the frequency as well as the site of stimulation [145–148], which can affect both intra and inter-population interactions. For instance, experimentally it was demonstrated that low-frequency electrical stimulation ($\leq$ 100 Hz) of thalamic structures can induce short-term synaptic facilitation, likely mediated by glutamate release [148, 149], whereas stimulation of BG nuclei was associated with short-term synaptic depression, likely mediated by local GABA release [145, 146]. This site-specificity may be lost during higher frequencies of electrical stimulation ($>$ 100 Hz) which are usually associated with suppression of neuronal activity due to short-term synaptic depression [148, 150, 151]. Yet, short-term or long-term synaptic plasticity alone may be insufficient to capture the whole range of stimulation-induced effects in modeling studies [148, 151]. Alternatively, incorporation of both short-term and long-term synaptic plasticity in models may provide a more accurate description of synapse-level effects vs. network-level effects of stimulation.

This paper is meant to serve as conceptual foundation of the time-shifted stimulation of sufficiently distant neuronal populations, applicable with different stimulation modalities. For direct electrical brain stimulation, e.g., DBS, one may additionally calculate the spatial distribution of the electric field to predict the volume of tissue activated (VTA), initially introduced by Rattay (1986) [152]. Of note, this approach considers stimulus effects on a silent axon [153] as opposed to intrinsically active neurons. For instance, VTA-based analysis [154–156] was used in combination with accurate DBS electrode location [157] and co-registration of patients' brains onto an average template brain [158] in retrospective group-level analyses in large patient cohorts to provide optimal stimulation targets or optimal connectivity patterns for the prediction of favorable outcomes in future patients [159]. However, while the co-registration of VTA and magnetic resonance imaging (MRI) provides a visual approximation, its validity critically depends on the underlying theoretical model, specifically ignoring intrinsic neuronal dynamics and local impedance changes [159]. The same principles and limitations apply to a VTA-based approach for the design of multi-contact leads for multichannel stimulation protocols [160]. Accordingly, more detailed models taking into account representation of the three-dimensional (3D) neuroanatomy, the time-dependent electric field generated by DBS electrodes, and the biophysical mechanisms that regulate the neural response to stimulation [161–164] may shed light on the mechanisms of action of electrical stimulation [165].

STDP-induced reorganization of cortical networks by stimulation has been addressed in a number of experimental works related to our computational results. For instance, paired stimulation [166], i.e., time-locked pairing of two stimulations delivered to two sites, allows to directly activate, e.g., cortico-cortical pathways connecting two areas, to study changes in inter-population connections through STDP [95, 167]. Experimental findings in humans suggest that STDP-like changes in cortico-cortical inter-hemispheric connectivity crucially depend on the delay (time shift) between paired stimuli delivered to two sites (also see Fig 9B) in the primary motor cortex [91, 92], as well as paired stimulation protocols targeting intra-hemispheric connectivity with short-range [93, 168] and long-range [93, 94] pairing delays (also see Fig 9A). Amazingly, bidirectional STDP-like changes in inter-hemispheric connections can be achieved by changing which hemisphere is stimulated first [169].

A recent *in vivo* study in awake, behaving monkeys demonstrated that STDP could be induced via time-shifted paired stimulation of two interacting cortical sites (namely, A and B) [95]. By assuming a fixed stimulation protocol and varying the delay (time shift) between stimuli, STDP-like changes in the strength of connections between the two sites were achieved

which were similar to classical single-cell findings *in vitro* [5–7]. For instance, positive delays between paired stimuli (i.e., pre-before-post or A→B) led to a strengthening of inter-population synapses, whereas negative delays (i.e., post-before-pre or B→A) weakened the synaptic connections [95]. Moreover, small time shifts between stimuli (< 20 ms) induced significant synaptic changes, whereas greater time shifts (> 50 ms) fell outside the functional temporal window of STDP [95], similar to the classical STDP profile [6, 7]. Interestingly, consistent with previous computational findings [96] and our results, a zero delay between paired stimulations did not induce significant changes in inter-population connections [95].

Ultimately, reorganization of cortical networks following stimulation [170–174], e.g., cortical remapping during perceptual performance, can be also interpreted as some sort of decoupling mediated by plasticity [175–177]. For instance, temporally correlated sensory inputs to the digits can lead to the merging of digit representations on the cortical surface, associated with topographic reorganization of the primary somatosensory cortex (SI) evidenced by electroencephalography (EEG), magnetoencephalography (MEG), and functional magnetic resonance imaging (fMRI) [170–174]. Particularly, it was shown that synchronous co-activation of the digits in humans led to an increase in temporal coherence of the fMRI signal due to temporal coincidence between the two-digit inputs, whereas asynchronous co-activation induced no significant change [172, 173]. Increased coherency is associated with reduced digit separation for the synchronous input, i.e., cortical representations for synchronously co-activated fingers moved closer together, whereas asynchronously co-activated fingers showed segregated cortical representations [172, 173]. Although mechanisms behind these changes are unclear, stimulation-induced strengthening/weakening of cortical synaptic connections via frequency-dependent synaptic plasticity may be one of the candidates that has been linked to the tactile discrimination performance [175–177].

Cognitive processes are related to structure-function relationships, in which the product of structurally and functionally interconnected brain areas is the basis for higher brain functions [16]. While we do not claim that our simple models might sufficiently explain complex cognitive processes, it may nevertheless contribute to testable predictions as to how cognitive processes may be impaired, e.g., by abnormal transmission delays. For instance, impaired cognitive function in multiple sclerosis (MS) may reflect damage to brain regions due to inflammatory demyelination, thus causing inhibition of axonal transmission. Potentially, impairments of cognitive (and non-cognitive) processes due to abnormal transmission delays have been found in MS [178]. Specifically, long-term memory, which is believed to be related to plastic changes of synapses, is one of the most consistently impaired cognitive functions in MS and is seen in 40–65% of patients [179].

Our study is meant to serve as a starting point for a top-down development for time-shifted stimulation protocols administered to remote targets. For comparison, for the improvement of DBS in PD, a top-down approach first used rather simple models, such as phase oscillator networks [72] and later more detailed and advanced models [130, 180, 181]. Remarkably, already the computational studies in simpler models, e.g., phase oscillator networks, provided key predictions, regarding long-lasting desynchronization [72, 131], cumulative desynchronization effects [73] and optimal stimulus patterns [79, 182] and parameters [183]. These computational studies revealed non-trivial predictions about stimulus-response characteristics of plastic neuronal networks, specifically the qualitative difference of acute effects (observed during stimulation), acute after-effects (shortly after cessation of stimulation) and long-term after-effects [72, 79, 140]. In addition, this top-down approach revealed fundamentally different effects of desynchronizing vs. decoupling stimulation patterns [80]. These findings and phenomena were substantially different from what was known about regular DBS. Furthermore, these computationally derived results served as predictions and were critical to the

development of appropriate experimental and study protocols, ultimately enabling to verify these predictions in animal experiments [74, 76–78, 184] and clinical studies [75, 185, 186].

We presented a simple and generic stimulation approach that causes pronounced changes of the network dynamics and connectivity, outlasting the cessation of stimulation. While the conclusions are mainly predictions, the model has the potential to advance clinical therapeutic techniques for a range of pathological conditions. Simple network models can be employed in order to thoroughly understand a wide range of stimulation-induced effects by performing detailed numerical and analytical analysis [72, 73, 80, 130, 187–189]. This enables us to make predictions that guide the analysis of more detailed and biophysically realistic networks [180, 188–193] as well as pre-clinical [74, 76, 194, 195] and clinical studies [75, 186, 196]. Although variations of model parameters may crucially determine collective dynamics and inter-population connectivity pattern, starting with such simple models and further refining them enables to generate testable predictions for numerical studies in more biophysically realistic models as well as for pre-clinical and clinical studies, e.g., by employing whole-brain modeling and inclusion of patient-specific connectivity patterns tuned to match electrophysiology and neuroimaging data [188, 189, 193].

The two-neuron motif is a simple, yet well-understood basic model in computational neuroscience [106, 108, 115, 116]. Results of the two-neuron motif served as predictions for the two-population model and were tested by numerical comparison with results obtained in the latter model. On the other hand, the two-population model is an established model for cortical neuronal networks with modular structure [84, 86, 99, 100]. It is widely used to study reliable signal propagation across cortex [84], inter-hemispheric phase coherence despite long-range transmission delays [100], synchronization between oscillations emerging from separated cortical areas [197], as well as *in vivo* conditioning protocols that produce cortical plasticity [95, 96]. In that sense, as demonstrated by our thorough analysis, the comparably simple two-neuron motif represents the dynamics of the two-population model convincingly well. Results obtained in the two-population model provide non-trivial predictions for experimental and clinical tests. The validity of these predictions will, in turn, depend on the validity of the two-population model. While the latter was thoroughly derived [84, 86, 99, 100], it cannot account for all possible additional features of cortical networks, e.g., ranging from layering to glial cells. The success of computational studies [96] to explain experimental outcomes of paired stimulation protocols that produce cortical plasticity [95] suggests that simple cortical models incorporating STDP may be able to explain the neural mechanisms underlying stimulation-induced cortical plasticity between two populations. Our results thus may contribute to the further optimization of a variety of bi-/multichannel stimulation protocols aimed at the therapeutic reshaping of brain networks.

## Supporting information

**S1 Fig. Balanced amplification in the inhibition-stabilized local network.** (**A**) Diagram of the isolated module M1 (with no projections from the module M2) as a balanced circuit with an excitatory (triangle) and an inhibitory (circle) population. Excitatory connections are green and inhibitory connections are red. (**B**) Raster plot (top) and population activity (middle) of the isolated module M1. (Bottom) Fourier transform frequency of the population activity (left) and distribution of the spike count irregularity (right) of the isolated module M1. (**C**) (Top) Firing rate of the excitatory ($r_E$) and inhibitory ($r_I$) populations in response to a pulse of input ($I_E$) to the excitatory population at time $t = 0$ s that sets $r_E = 10$ Hz ($r_I = 0$ Hz; representing the baseline of activity). (Bottom) Excitatory-inhibitory response ratio for the top panel.
(EPS)

**S2 Fig. Distribution of the intra- and inter-population synaptic strengths in the model.** (**A,B**) The intra-population inhibitory synaptic strengths were picked from a Gaussian distribution with mean 0.8 and standard deviation 0.05, whereas the intra-population excitatory synaptic strengths were picked from a Gaussian distribution with mean 0.2 and the same standard deviation. (**C**) The inter-population excitatory synaptic strengths were chosen from a Gaussian distribution with mean 0.2 and standard deviation 0.05.
(EPS)

**S3 Fig. Suppression of the synaptic strengths between two neurons by time-shifted stimulation in the absence of delays.** Time course of neuronal activity (N1/N2) and the synaptic strengths ($g_{21}/g_{12}$) are shown for two neurons with STDP parameters $A_+ = 0.008$, $A_- = 0.005$, $\tau_+ = 10$ ms and $\tau_- = 20$ ms, and $|\xi| = 0.0$ ms. Stimulation parameters were ($\Delta t$, $T$) = (15, 30) ms.
(EPS)

**S4 Fig. Decoupling by time-shifted stimulation in the network in the absence of delays.** (**A1,A2**) Raster plots and population activities are shown for the modules (M1/M2) before/after stimulation on/offset. (**B,C**) Distribution of the pairwise correlation and spike count irregularity of each module before (grey) and after (colored) stimulation. (**D**) Snapshot of phase lags ($\Delta\phi$) between synchronous discharges from the two modules before ($t = 9$ s, left) and after ($t = 25$ s, right) stimulation. (**E**) Fourier transform frequency of the population activity of each module before (grey) and after (colored) stimulation. (**F**) Time course of the inter-population mean coupling ($G_{21}/G_{12}$). (**G**) Distribution of the inter-population synaptic strengths before (grey) and after (colored) stimulation. (**H**) Time course of the firing rate of neurons ($f1/f2$) in each module. The modules were stimulated with time shift $\Delta t = 15$ ms and frequency $\nu = 33.3$ Hz for the duration of $T_{\text{stim}} = 5$ s (highlighted area in F and H). STDP parameters were $A_+ = 0.008$, $A_- = 0.005$, $\tau_+ = 10$ ms and $\tau_- = 20$ ms.
(EPS)

**S5 Fig. Initial and final network topology.** (**A1**) Structural connectivity matrix of the network (i.e., inter-population excitatory-to-excitatory connections) drawn using a binary representation, i.e., connectivity matrix array, $c_{ij} = 1$ if the two neurons are connected, and $c_{ij} = 0$, otherwise. There were $N_{\text{ex}} = 160$ excitatory neurons within each population. (**A2**) Degree distribution of the initial network topology with random connection probability $p_{\text{inter}} = 0.15$. (**B1**) Final synaptic strength matrix of the network. (**B2**) Binary representation of the final connectivity matrix of the network constructed by introducing a threshold ($h = 0.3$), i.e., connectivity matrix array, $c_{ij} = 1$ if $g_{ij} \geq h$, and $c_{ij} = 0$, otherwise. (**B3**) Degree distribution of the final network topology. Panels B1-B3 were depicted for the STDP parameters used in Fig 5Ba with ($A_+/A_-$, $\tau_+/\tau_-$) = (1.6, 0.5). (**C1-C3**) Same as B1-B3, but for the STDP parameters used in Fig 5BC with ($A_+/A_-$, $\tau_+/\tau_-$) = (1.6, 1.6).
(EPS)

## Author Contributions

**Conceptualization:** Mojtaba Madadi Asl, Alireza Valizadeh, Peter A. Tass.

**Formal analysis:** Mojtaba Madadi Asl, Alireza Valizadeh, Peter A. Tass.

**Investigation:** Mojtaba Madadi Asl.

**Methodology:** Mojtaba Madadi Asl, Alireza Valizadeh, Peter A. Tass.

**Project administration:** Alireza Valizadeh.

**Supervision:** Alireza Valizadeh, Peter A. Tass.

**Visualization:** Mojtaba Madadi Asl.

**Writing – original draft:** Mojtaba Madadi Asl, Alireza Valizadeh, Peter A. Tass.

**Writing – review & editing:** Mojtaba Madadi Asl, Alireza Valizadeh, Peter A. Tass.

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
