## [Decision Letter · Decision Letter 0]

16 Aug 2022

Dear Dr. Madadi Asl,

Thank you very much for submitting your manuscript "Decoupling of interacting neuronal populations by time-shifted stimulation through spike-timing-dependent plasticity" for consideration at PLOS Computational Biology.

As with all papers reviewed by the journal, your manuscript was reviewed by members of the editorial board and by several independent reviewers. In light of the reviews (below this email), we would like to invite the resubmission of a significantly-revised version that takes into account the reviewers' comments.

Please be sure in particular to provide clear reasoning/justification for all model design decisions and parameter settings. Please also be sure to clarify that you will share the code in the case of acceptance, per the PLOS Computational Biology guidelines (https://journals.plos.org/ploscompbiol/s/code-availability).

We cannot make any decision about publication until we have seen the revised manuscript and your response to the reviewers' comments. Your revised manuscript is also likely to be sent to reviewers for further evaluation.

Sincerely,

Blake A Richards

Academic Editor

PLOS Computational Biology

Daniele Marinazzo

Section Editor

PLOS Computational Biology

Please be sure in particular to provide clear reasoning/justification for all model design decisions and parameter settings.

Reviewer's Responses to Questions

**Comments to the Authors:**

Reviewer #1: The authors using model with channels and insert STDP. The authors are able to illustrate that subtle changes to a stimulation protocol can have a significant impact.

I believe that the great differential of this work is in the introduction of a time change between the stimuli delivered to two populations of interacting neurons, which can effectively dissociate them. But this should be clearer in the manuscript.

I have only a few indications and later it can be recommended for publication

1) According to the text: the migration of populations was based on initially strong interpopulation links. I suggest that the authors justify this possibility in real human brains.

2) Could the authors estimate a final network topology? How? And if possible, what is it for the conditions addressed in this work?

3) Do the authors test the thresholds of synaptic weights so that the collective behavior of the model does not lose its proximity to the realistic situation?

4) How were the model and conclusions of this work validated?

5) This work can be used to explain cognitive processes? How?

6) Recommendations for reading

Motifs in Brains Networks. PLoS Biol. 2004 Nov; 2(11): e369.

Spike timing-dependent plasticity induces non-trivial topology in the Brain. Neural Networks 88 (2017) 58–64.

Effects of the spike timing-dependent plasticity on the synchronization in a random Hodgkin–Huxley neuronal network. Commun Nonlinear Sci Numer Simulat 34 (2016) 12–22.

The human nervous system: Structure and function. https://doi.org/10.1007/978-1-59259-730-7

Reviewer #2: Summary

In this paper, the authors present a computational model of 2 interacting cells/ populations to demonstrate that a) time-shifted stimuli can cause desynchronization and decoupling, and b) that the inclusion of delays into the model shape the model predictions.

It should not be surprising that when the temporal dynamics of the neural units are adjusted so that they violate the LTP conditions of STDP (or conversely, satisfy the conditions of LTD) then the connection strengths between those units are reduced. However, what is novel is the inclusion of realistic delays and some effort to provide predictions that can have immediate utility in clinical settings and I believe that to be the true value of this paper.

What is surprising is that the effect of desynchronization and decoupling persists after the stimulation is removed – and it is this point that is at the core of this model’s translational potential and importance.

This paper is extremely well written and is logically structured. One concern is that the model predictions are highly sensitive to the parameters used and not all have been justified or explained (or explored, understandably). I feel that more careful tying of the model to physiological parameters would make this paper stronger.

Abstract and Introduction

There is a slight disconnect in the introduction in that it is focussed almost entirely on PD as an exemplary clinical case where desynchronization of neural circuits has therapeutic potential. However, the model presented is a generalised case that is not ‘tuned’ to represent any particular region/pathology.

I think that the authors should consider ether: a) presenting this as a generalised model and place it in the context of a broader range of pathologies/examples OR b) make the effort to tune the model itself to represent regions implicated in PD specifically.

The authors mention the issue of ‘pathological connectivity’ throughout the paper but I think this could be addressed with more precision. For example, the paper should include a more detailed discussion about the physiological nature of ‘increased/ pathological connectivity’ or ‘strong synaptic connections’ in PD (or indeed any other relevant pathologies). Which neuron types are involved/ changes in neural density/ synapses/ receptors/ etc. not to change the model as of course a model is a simplification of the biology, but so that the reader has a better understanding of the physiology in this case and therefore how the model relates.

Methods

It would be helpful to list the parameters used (with descriptions and references) so it is easy to see what is changed for each simulation. Because the simulations are not listed in methods (which is fine) I had to go back and forth to understand which parameters were changed for each figure and this would be solved with a simple table.

A is used twice as the adjacency matrix and also as a measure of network activity – the authors may wish to change one of these to reduce reader confusion.

The temporal parameters of STDP are very fast (10/20 ms) – how have these been determined? I realise there is some exploration of this space in Fig. 5 – but even here these are constrained to be in the ms range which is fast for true LTP/LTD mechanisms.

What is the external stimulation intended to represent? I agree that modelling it as a simple input current is a sensible first step but there are a few considerations: if it is intended to model tDCS/tACS then this affects E/I cells differently, if it is a sensory stimulus then the authors should justify that it targets E/I cells equally as this is not always thought to be the case (for all stim types).

How were the inter-module delays chosen/ which tracts do they represent?

Do the initial conditions for G/g have any effect on the model predictions?

How is the noisy input determined/ what are the parameters?

Results

Below I include my interpretation of each result in case I have misunderstood followed by any questions/comments. NB there is some repetition with comments from methods.

Fig.2 – pairwise connectivity is shown for the 2-neuron model for a range of delta_t (delays) and T (inter-pulse interval or intra-burst freq). We see that for increasing delays we have more complex patterns of the phase-space (including a desynchronised state for delta_t = 0 and Xi>0).

Fig. 3 – time course of the spikes and the connections in 2-neuron model. Firing rates are equivalent for changes in delay indicating this doesn’t underlie the changes in synchrony.

Fig. 4 – Raster plots to sow the anti-correlated firing before stim onset within the modules (does the output change if they are correlated?) and then uncorrelated after desynchronization. Bar charts show that correlation decreases while irregularity increases. Spike rate reduces, coupling strength reduces. The persistent effect of the stimulation on desynchronization and decoupling is of great clinical interest. It appears to be driven by lower firing rates post-stimulation. In this case its important to know under what conditions the firing rate remains low after stimulation ceases – for example, the noise level and the local connectivity strengths (E and I) will play a role in this. A fuller discussion of and justification for the values used would be helpful.

Fig. 5 – Examines the effects of the STDP parameters on the model behaviour. It seems that the model parameters have a significant effect on the model output. I wonder if the parameters can be tuned to any biological ones so that the reader can get a sense of how these results translate to the brain? I could not see any justification for the choices for tau_+ and tau_- (I assume the learning rates are arbitrary but perhaps these could also be tuned to match empirical data). How have these values been set and are they realistic? LTP /LTD act on longer timescales than this. I think this point is important if these results are to be translated which I assume to be the authors’ aim. How is the threshold calculated as it seems to change in each simulation – or is it just an observed value?

Fig. 6 – For Xi fixed (=10) shows the effect of continuous vs burst stim on model predictions and demonstrates that continuous has a more profound effect than burst. This is explained by the fact that there is no STIM-OFF period when the connections start to increase again. In reality, however, the continuous stim would perhaps invoke slower secondary processes such as adaptation which would compensate somewhat for this effect. A suitable control is demonstrated by using synchronous stimulus. Perhaps I misunderstood but for Xi=10 and delta_t=0, then the model should desynchronise according to Fig. 2? (the parameters would correspond to the point just left of point b in Fig. 2 C). The authors mention jittering at delta_t=0 but this isn’t reflected in Fig. 2.

Fig. 7 – For the 2 cases Xi=0, Xi=10 ms then we see the effect of varying delta_t and T (or v) on fano factor and averaged coupling. We get some specific predictions here – for example when Xi=10 delta_t must be between <10 ms and v > 60 Hz or <30 Hz in order to get desynchronization and decoupling. However slight deviations from these parameters produce undesirable effects so prediction precision will be important for translation. This is an important figure.

Discussion

Limitations

The authors have discussed the limitations of the model satisfactorily.

Advance in the field

This paper has the potential to advance clinical therapeutic techniques for a range of pathological conditions. In order for this to be the case the authors should try to link the model more closely with the physiology of a specific brain region/regions by justifying the parameters more carefully. It seems clear that the model is highly sensitive to parameter choices and so it’s unrealistic to expect direct translation of the model to clinical use but the authors could address next steps in this journey (whole-brain modelling, inclusion of patient specific connectivity patterns, tuning to match EEG spectra, as examples).

Discussion of relevant literature

There is a good discussion of pathological application of this kind of stimuli. However there is also a body of literature (Dinse, Godde, Vidyasagar) that have looked at the effect of asynchronous inputs to the somatosensory system within a single hemisphere (via digit stimulation) and reported various measurements that can be interpreted as ‘decoupling’ (for example cortical remapping, task performance). I think including these can only strengthen the paper as they are not hypothetical applications, but real ones.

Reviewer #3: In this paper, authors utilized a tractable modeling framework consisting of LIF neurons to theoretically analyze stimulation patterns required to de-synchronize pathological pathways in cortical regions. Although it was not mentioned which brain regions have such potential to be simultaneously stimulated, authors used their methodology to formulate suppression of strong inter-population synaptic connectivity through spike-timing dependent plasticity (STDP). Authors showed that multi-site stimulation parameters can be controlled through their model by incorporating impacts of STDP.

I should clarify that the problem that authors tried to address is very important and timely, and this is why I would like to support this research. A modeling approach is required to tackle such a problem. These two aspects attract my attention. However, I have some big problems with the approach and the problem design that I explain below. Before getting into this point, I should also ask from authors why they have not tried to validate their modeling – probably partially – with some existing neural recordings (like local field potentials) during (or after) deep brain stimulation (although the DBS pulses are delivered to a single population, I think some aspects of the modeling could be validated with experimental data). This is an important point for the development of translational methods. Back to modeling framework, I would like to raise the following points:

1) Although I liked the way the two interacting population of neurons were formulated, why the model is in exact support of authors theoretical hypothesis?, i.e., the oscillations were driven only by synaptic connectivity between two populations and other effects like neuronal activities within each sub-population and background synaptic activities have no effect.

2) What are examples of such neuronal activities/oscillations in some brain regions that authors can provide to support their modeling design?

3) The assumption of two interacting excitatory population, the Hebbian type network, is very simplistic. There are several significant theoretical works in support of inhibitory stabilizing networks and balanced amplification that hugely challenge such simple network modeling. Some works by Kenneth D Miller (as the senior author) in 2009 and 2015 can be considered.

4) Why the role of short-term synaptic plasticity was not incorporated, and more importantly, not discussed. Stimulation-induced impact of the short-term synaptic plasticity can have very different effects on glutamatergic and GABAergic synapses, which can affect both inter and intra population interactions.

5) The impact of electrical stimulation was considered with very minimal biophysical constraints. Various studies by McIntyre and colleagues addressed this point.

As a modeler, I totally understand that we cannot expect that a model provides all such details. However, in the absence of experimental data, a theoretical study should address various aspects that a model can offer, rationales for ignoring some constraints, and discussions on the limitation of the model. The question raised in this paper is really an interesting one. I think a more appropriate methodology should be designed.

**Have the authors made all data and (if applicable) computational code underlying the findings in their manuscript fully available?**

Reviewer #1: **No: **

Reviewer #2: **No: **I couldn't see the code anywhere.

Reviewer #3: **No: **I could not find the code of this paper withing the main body of the article.

PLOS authors have the option to publish the peer review history of their article (what does this mean?). If published, this will include your full peer review and any attached files.

Reviewer #1: No

Reviewer #2: No

Reviewer #3: **Yes: **Milad Lankarany
---

## [Decision Letter · Decision Letter 1]

5 Jan 2023

Dear Dr. Madadi Asl,

We are pleased to inform you that your manuscript 'Decoupling of interacting neuronal populations by time-shifted stimulation through spike-timing-dependent plasticity' has been provisionally accepted for publication in PLOS Computational Biology.

Before your manuscript can be formally accepted you will need to complete some formatting changes, which you will receive in a follow up email. A member of our team will be in touch with a set of requests. Please also be sure to fix the remaining typos identified by Reviewer 2.

Best regards,

Blake A Richards

Academic Editor

PLOS Computational Biology

Daniele Marinazzo

Section Editor

PLOS Computational Biology

Reviewer's Responses to Questions

**Comments to the Authors:**

Reviewer #2: The authors have addressed all of my points satisfactorily. I appreciate the efforts made to properly justify the parameters and broaden the Introduction to place the model more generally in the field.

I identified the following minor typos that the authors may wish to address but otherwise I have no further comments.

1. Line 562 – close bracket without an opening one.

2. Line 577 type.

3. Line 580 ‘a similar outcome’/’similar outcomes’.

4. ‘Hebbian-type network’ – Hebbian refers to the connections not the network (as far as I understand).

5. Line 775 differentially?

6. Line 879 /887 enabling/enables us to...

Reviewer #3: The authors addressed all points, Great job.

**Have the authors made all data and (if applicable) computational code underlying the findings in their manuscript fully available?**

Reviewer #2: Yes

Reviewer #3: Yes

PLOS authors have the option to publish the peer review history of their article (what does this mean?). If published, this will include your full peer review and any attached files.

Reviewer #2: No

Reviewer #3: **Yes: **Milad Lankarany

---

## [Editor Report · Acceptance letter]

16 Jan 2023

PCOMPBIOL-D-22-01018R1 

Decoupling of interacting neuronal populations by time-shifted stimulation through spike-timing-dependent plasticity

Dear Dr Madadi Asl,

I am pleased to inform you that your manuscript has been formally accepted for publication in PLOS Computational Biology. Your manuscript is now with our production department and you will be notified of the publication date in due course.

With kind regards,

Zsofi Zombor
